# SUMOylation of Rho-associated protein kinase 2 induces goblet cell metaplasia in allergic airways

Dan Tan [1,2,7], Meiping Lu [3,7] ✉, Yuqing Cai[3], Weibo Qi[4], Fugen Wu[5], Hangyang Bao [1], Meiyu Qv [1], Qiangqiang He [1], Yana Xu [1], Xiangzhi Wang[3], Tingyu Shen[1], Jiahao Luo[1], Yangxun He[1], Junsong Wu[6], Lanfang Tang[3], Muhammad Qasim Barkat [1], Chengyun Xu [1,2,3] ✉ & Ximei Wu [1,2] ✉

Allergic asthma is characterized by goblet cell metaplasia and subsequent mucus hypersecretion that contribute to the morbidity and mortality of this disease. Here, we explore the potential role and underlying mechanism of protein SUMOylation-mediated goblet cell metaplasia. The components of SUMOylaion machinery are specifically expressed in healthy human bronchial epithelia and robustly upregulated in bronchial epithelia of patients or mouse models with allergic asthma. Intratracheal suppression of SUMOylation by 2-D08 robustly attenuates not only allergen-induced airway inflammation, goblet cell metaplasia, and hyperreactivity, but IL-13-induced goblet cell metaplasia. Phosphoproteomics and biochemical analyses reveal SUMOylation on K1007 activates ROCK2, a master regulator of goblet cell metaplasia, by facilitating its binding to and activation by RhoA, and an E3 ligase PIAS1 is responsible for SUMOylation on K1007. As a result, knockdown of PIAS1 in bronchial epithelia inactivates ROCK2 to attenuate IL-13-induced goblet cell metaplasia, and bronchial epithelial knock-in of ROCK2(K1007R) consistently inactivates ROCK2 to alleviate not only allergen-induced airway inflammation, goblet cell metaplasia, and hyperreactivity, but IL-13-induced goblet cell metaplasia. Together, SUMOylation-mediated ROCK2 activation is an integral component of Rho/ROCK signaling in regulating the pathological conditions of asthma and thus SUMOylation is an additional target for the therapeutic intervention of this disease.

Asthma is one of the most prevalent chronic respiratory diseases worldwide[1,2]. The cardinal features of asthmatic airway remodeling consist of goblet cell metaplasia of airway epithelia, mucus hypersecretion, thickening of the basement membrane, increases in vascularity, smooth muscle cell hypertrophy and hyperplasia[3]. Goblet cell metaplasia of airway epithelia is defined as the transformation of airway epithelia to mucous cells and characterized by mucous hypersecretion that is associated with the morbidity and mortality of asthma pathological conditions[4,5]. Goblet cell metaplasia of allergic airway is thought to be resulted from the transition of Club cells and ciliated cells, which are often assigned to multiple inflammatory stimuli. Among a variety of Th2 cytokines, interleukin-13 (IL-13) is believed to be extremely critical for goblet cell metaplasia. Intratracheal instillation of IL-13 alone into mouse airway is sufficient to induce airway goblet cell metaplasia[6], while elimination of IL-13 is capable of reversing the

metaplastic goblet cells into ciliated epithelial cells in airway epithelial cell culture[7].

Rho family of small GTPases, including RhoA, RhoB, and RhoC, function as molecular switches that turn on/off the signal transduction pathways in response to various stimuli[8,9]. The three closely related Rho GTPases can interact with the same downstream effectors, such as Rho-associated protein kinase 1/2 (ROCK1/2), but have substantially distinct roles in the regulation of various biological processes[10,11]. ROCK1 and ROCK2 are highly homologous with 64% identical amino acid sequences[12]. RhoA-ROCK signaling pathway is believed to be involved in regulating allergic airway inflammation, remodeling, and hyperreactivity (AHR)[13], and is regarded as a potential therapeutic target of this disease[14–16]. In an ovalbumin (OVA) repeatedly challenged mouse model of allergic airways disease, haploinsufficiency of either ROCK1 or ROCK2 partially or completely abrogates airway hyper-reactivity, respectively, likely through their direct effects on smooth muscle cell and effects on mast cell degranulation, however, ROCK2 but not ROCK1 participates in the goblet cell hyperplasia of this mouse model of allergic asthma[17].

Small ubiquitin-related modifiers (SUMOs) are ubiquitin-like polypeptides that are covalently conjugated to the lysine residues of target proteins in a manner similar to ubiquitination. Both SUMOylation and deSUMOylation are dynamic and reversible post-translational processes and implicate in protein localization, stability and interaction[18,19]. SUMOylation occurs through a series of enzyme-catalyzed reactions mediated by a heterodimeric E1 SUMO-activating enzyme (SAE) 1 and 2, a unique SUMO E2 conjugating enzyme (UBC9), and a few SUMO E3 ligases, such as protein inhibitor of activated STAT (PIAS) family E3 ligases. Despite the fact that UBC9 is sufficient to transfer SUMO into substrates in vitro, specific E3 ligases are most often required for an efficient in vivo reaction[20]. In contrast, deSU-MOylation is enzymatically catalyzed by sentrin/SUMO-specific proteases (SENPs) in mammals[21,22]. A recent transcriptomic analysis of biopsies identifies patients with severe asthma have significant enrichment for SUMOylation, as compared to healthy control, despite high-dose inhaled corticosteroids[23].

In the present study, we attempted to determine the role of SUMOylation in the regulation of allergic airway goblet cell metaplasia and mucus hypersecretion and found that enrichment of SUMOylation in allergic airway epithelia SUMOylates and thereby activates ROCK2 to promote goblet cell metaplasia.

## Results

### SUMOylation machinery components are upregulated in allergic airway epithelia

To determine the expression patterns of SUMOylation machinery components in healthy bronchi, we surgically obtained the lobar bronchi from patients, who had peripheral lung cancers and underwent pulmonary lobectomy, and performed the immunohistochemistry staining for SAE1/2 and UBC9. Both SAE1/2- and UBC9- derived immunosignals were specifically and robustly detected in the human bronchial epithelia but not in bronchial smooth muscle (Fig. 1a). To investigate the expression patterns of SUMOylation machinery components in allergic airway epithelia, we obtained the bronchoalveolar lavage fluids (BALFs) from children with allergic asthma or foreign body aspiration (FBA) and harvested the cell pellets for immunostaining. Clara cell 10 kDa protein (CC10)-, SAE1-, SAE2-, and UBC9-derived immunosignals were robustly detectable, and the apparent overlapping immunosignals from CC10 and SUMOylation machinery components were readily observed in the BALF cells of children with asthma but not in those with FBA (Fig. 1b, c). To confirm the expression patterns of SUMOylation machinery components in allergic airway epithelia, we established an OVA-sensitized and -challenged mouse model with allergic asthma. Immunohistochemistry analyses consistently indicated that SAE1/2 and UBC9 were robustly upregulated in

the airway epithelia of OVA-challenged mice versus normal saline (NS)-challenged mice (Fig. 1d, e), this high expression of SAE1/2 and UBC9 in OVA-challenged lungs was confirmed further by quantitative RT-PCR (qPCR) and western analyses (Fig. 1f–h). In contrast, analysis of the mRNA levels of deSUMOylation enzymes indicated SUMO specific peptidase (SENP) 1, 6 and SENP2, 3, 5, 7 were downregulated or unchanged in OVA-challenged lungs as compared to in NS-challenged lungs, respectively (Supplementary Fig. 1). Thus, components of SUMOylation and deSUMOylation machineries were upregulated and downregulated in allergic airway epithelia, respectively, suggesting the potential importance of protein SUMOylation in regulating allergic airway epithelial remodeling.

### Intratracheal suppression of SUMOylation attenuates allergen-induced goblet cell metaplasia

To investigate the potential role of SUMOylation in the regulation of epithelial remodeling of allergic airways, we intratracheally instilled the 2-D08, a small-molecule inhibitor of UBC9, at 10 or 30 mg/kg (50 µl/mouse), into mice at 2 h post each OVA or house dust mite (HDM) atomization (Fig. 2a and Supplementary Fig. 2a). Cell counting and classification of the BALF cells indicated the total inflammatory cell numbers in BALFs were significantly increased after OVA or HDM challenge, among macrophages (Mac), lymphocytes (Lym), eosinophils (Eos), and neutrophils (Neu), eosinophils were most strikingly increased (Fig. 2b, c and Supplementary Fig. 2b, c). 2-D08 significantly attenuated the number of inflammatory cells, including macrophages and eosinophils, and their infiltration into peribronchial and perivascular connective tissues of OVA- or HDM-atomized mice in a dose-dependent manner (Fig. 2b, c and Supplementary Fig. 2b, c). In addition, OVA or HDM atomization robustly increased the stained areas of Periodic Acid-Schiff (PAS) as well as Muc5AC in bronchial epithelia, whereas 2-D08 dose-dependently decreases OVA- or HDM-induced PAS- and Muc5AC-stained areas in bronchial epithelia (Fig. 2d, e and Supplementary Fig. 2d, e). Finally, spirometry results indicated OVA challenge significantly increased airway resistance in response to methacholine ranging from 0.025 to 0.8 mg/kg, whereas 2-D08 dose-dependently decreased airway resistance in response to methacholine ranging from 0.05 to 0.8 mg/kg (Fig. 2f). Likewise, HDM challenge robustly increased the airway resistance in response to methacholine ranging from 0.05 to 0.8 mg/kg, while intratracheal administration of 2-D08 significantly reduced HDM-induced the airway resistance in response to methacholine ranging from 0.1 to 0.8 mg/kg, respectively (Supplementary Fig. 2f). Thus, intratracheal inhibition of protein SUMOylation attenuated the airway inflammation, hyperreactivity, and goblet cell metaplasia in response to allergen stimulation.

Previous studies demonstrated that direct instillation of IL-13 into airway epithelia was sufficient to induce the mucus metaplasia and AHR, but not the bronchial inflammation[24,25]. IL-13 itself did not affect the mRNA and protein levels of components of SUMOylation machinery in human bronchial epithelial cells, 16HBE cells (Supplementary Fig. 3a, b). To rule out the possibility that 2-D08 attenuating the goblet cell metaplasia was secondary to its inhibitory effects on inflammatory response in the OVA- or HDM-induced asthma mouse model, we intratracheally instilled mice with bioactive recombinant mouse IL-13 (12 µg/mouse) to induce airway epithelial metaplasia, following intratracheal instillation of 2-D08 (Fig. 2g). IL-13 induced a significant increase in PAS and Muc5AC staining of bronchial epithelia, while 2-D08 at 10 and 30 mg/kg decreased the IL-13-induced PAS and Muc5AC staining in a dose-dependent manner (Fig. 2h, i). Thus, intratracheal suppression of protein SUMOylation directly reduced the airway goblet cell metaplasia in response to IL-13 stimulation.

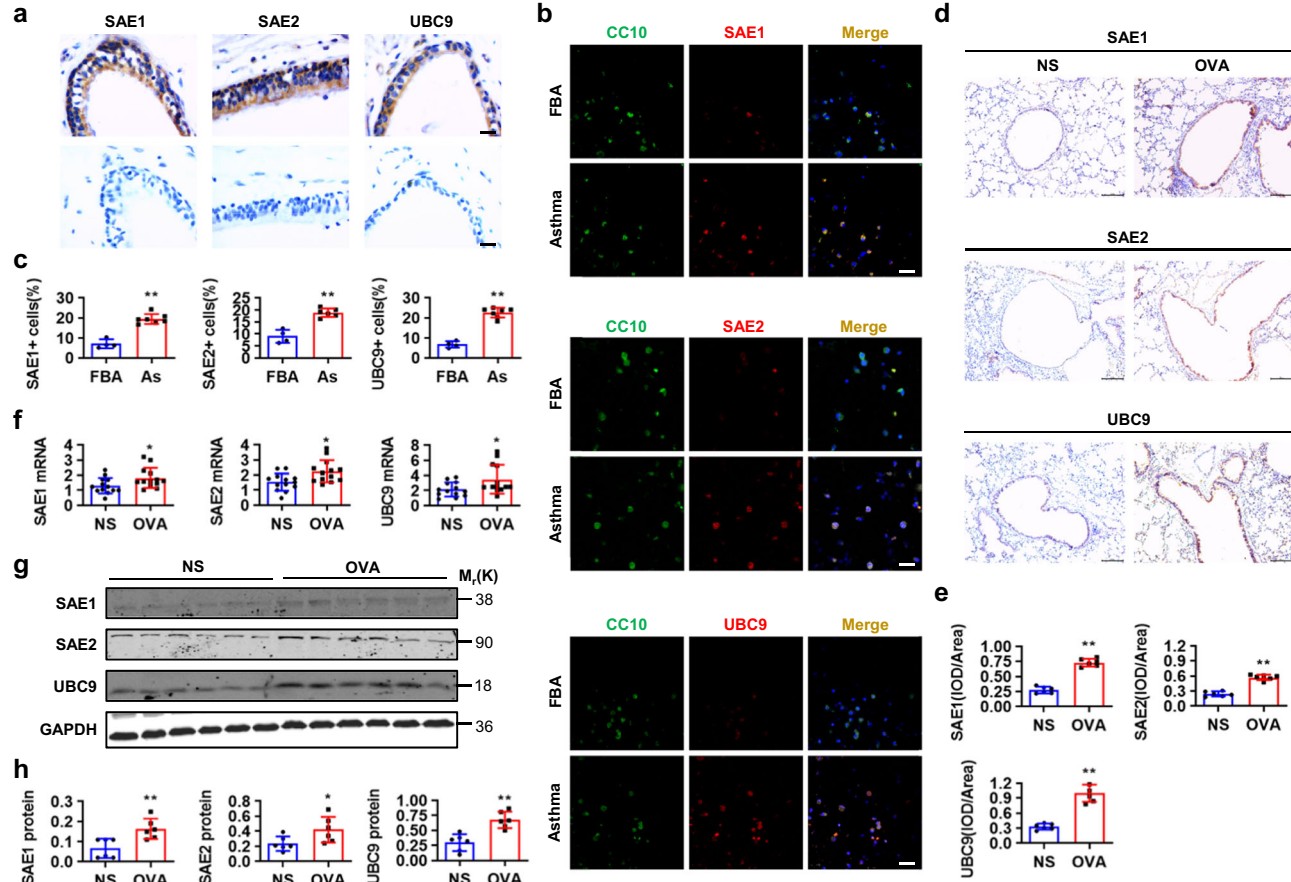

**Fig. 1 | Enriched SUMOylation in allergic airway epithelia.**
**a** Immunohistochemistry staining (IHC) for SAE1, SAE2 and UBC9 in human healthy bronchi ($n = 4$, upper: experimental group, bottom: negative control group, scale bar, 2 μm). **b, c** Immunostaining for CC10, SAE1, SAE2, UBC9, and DAPI in BALF cells from children with FBA ($n = 4$) or asthma (As, $n = 7$) and semi-quantifying the numbers of SAE1[+]-, SAE2[+]- or UBC9[+]-cells in CC10[+] cells (P values: <0.0001, <0.0001, <0.0001). Scale bar, 10 μm. **d–h** OVA-sensitized mice (each $n = 13$) were atomized

with 1% OVA or an equal volume of normal saline (NS) for 30 min, once daily for 7 d. Lungs were then subjected to paraffin-embedded sectioning, IHC, and semi-quantification (**d, e**, P values: <0.0001, <0.0001, <0.0001, scale bar, 10 μm), and to RNA isolation and qPCR (**f**, P values: 0.0254, 0.0109, 0.0345), bronchi were used for western analyses and semi-quantification (**g, h**, P values: 0.0057, 0.0373, 0.0009). Mean ± SD, unpaired two-tailed Student's t test, *$P < 0.05$, **$P < 0.01$. Source data are provided as a Source Data file.

## Suppression of SUMOylation inactivates ROCK2

To investigate the potential SUMOylation substrates controlling allergic airway goblet cell metaplasia, we performed an unbiased quantitative phosphoproteomics analysis in 16HBE cells treated with or without IL-13 and 2-D08. A total of 56,211 phosphorylation sites for 5283 phospho-proteins were identified and quantified, and proteins with a fold change ≥2.0 and ≤0.5 with $P < 0.05$ were considered significantly differential expression (Supplementary Data 1). Among them, 3264 phosphorylation sites were significantly upregulated after IL-13 treatment, whereas 625 IL-13-induced phosphorylation sites were significantly downregulated after 2-D08 treatment (Fig. 3a and Supplementary Data 1). Domain analysis highlighted both Rho guanine nucleotide exchange factors (RhoGEF) and GTPase-activating proteins (RhoGAP) were most tightly involved in the 2-D08 negating IL-13-induced phosphorylation of target proteins (Fig. 3b and Supplementary Data 2). In addition, of these proteins, ROCK2 exhibited to be one of the most modified proteins with multiple upregulated phosphorylation sites upon IL-13 stimulation (Fig. 3c). Western analyses confirmed that IL-13 induced the phosphorylation of ROCK2 at Ser1366 (p-ROCK2) in both the dose- and time-dependent manners (Supplementary Fig. 3c), whereas 2-D08 treatments dose-dependently reduced the IL-13-induced p-ROCK2 and phosphorylated myosin light chain 2 (p-MLC2), a downstream effector of p-ROCK2, in 16HBE cells (Fig. 3d, e). Moreover,

immunofluorescence staining and western analyses indicated intratracheal instillation of IL-13 robustly increased the p-ROCK2 levels in bronchial epithelia and bronchi of mice, respectively, and this effect of IL-13 was largely or completely abolished by intratracheal instillation of 2-D08 at 10 or 30 mg/kg, respectively (Fig. 3f–i). Finally, immunofluorescence staining and western analyses consistently revealed that 2-D08 at both 10 and 30 mg/kg robustly diminished OVA- or HDM-induced p-ROCK2 in bronchial epithelia and bronchi of mice, respectively (Fig. 3j–m and Supplementary Fig. 2g–i). Thus, intratracheal suppression of SUMOylation inactivated ROCK2 in bronchial epithelia in response to allergen or IL-13 stimulation in mice.

## Suppression of SUMOylation attenuates goblet cell metaplasia in response to constitutive activation of RhoA in bronchial epithelia

To determine whether activation of RhoA was sufficient to induce airway goblet cell metaplasia and inhibition of SUMOylation attenuated RhoA-induced ROCK2 activation and goblet cell metaplasia, we crossed conditional constitutively active form of RhoA knock-in (*caRhoA^+/−*) mice with *CC10-Cre^ERT2* mice to generate *CC10-Cre^ERT2; caRhoA^+/−* and age- and sex-matched *CC10-Cre^ERT2* control littermates at 8 weeks of age. Tamoxifen at 200 mg/kg was intraperitoneally injected on day 0, 1, 2, 3, 4 to induce the expression of *caRhoA* in airway epithelia, 2-D08 at 10 and 30 mg/kg was

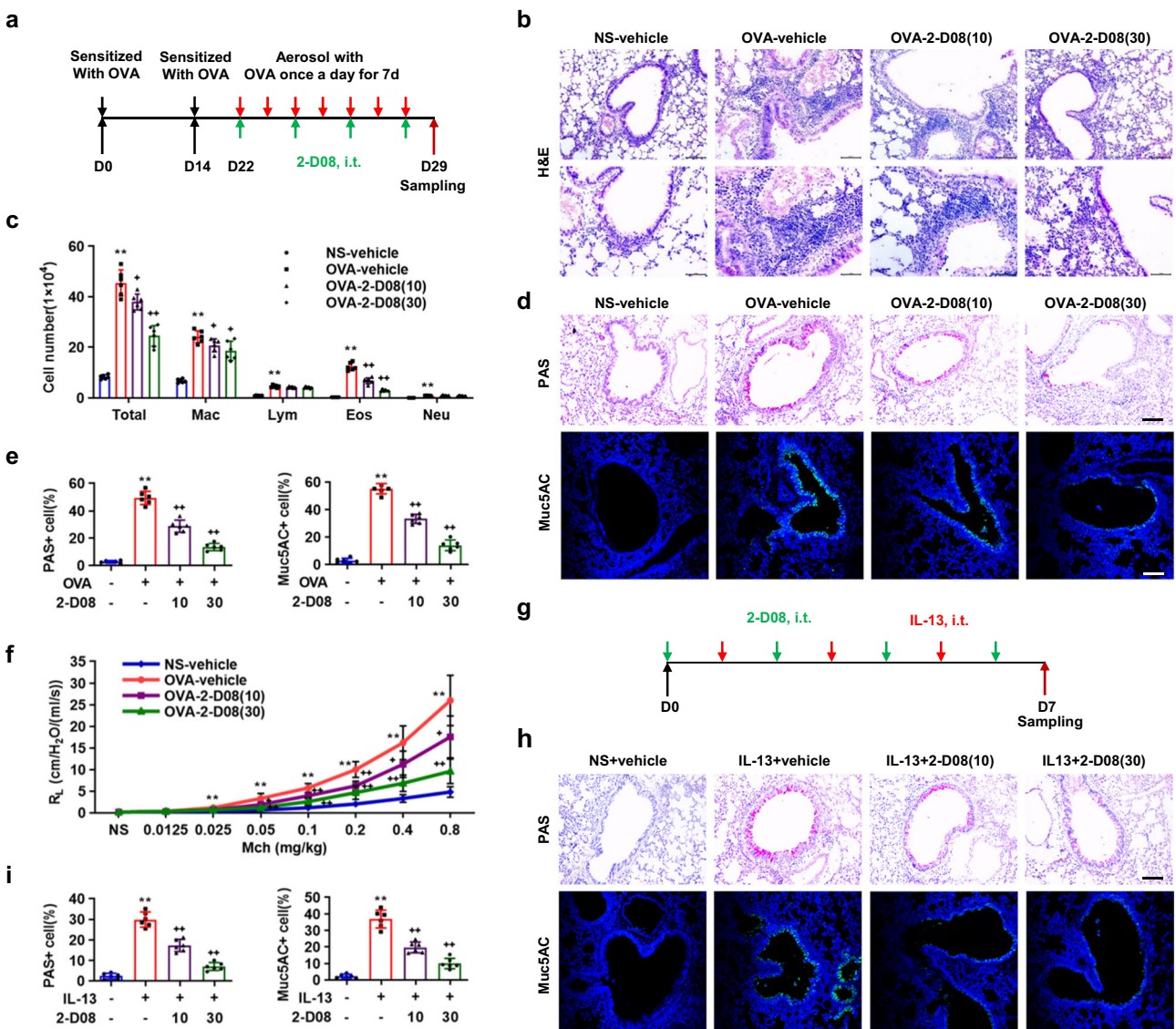

**Fig. 2 | Suppression of SUMOylation attenuates OVA- or IL-13-induced goblet cell metaplasia. a** OVA-sensitized mice were atomized with 1% OVA or an equal volume of NS for 30 min, once daily for 7 d. 2 h after each OVA atomization, mice were intratracheally administered with either 50 μl/mouse of 2-D08 at the dosages of 10 and 30 mg/kg or vehicle. Scale bar, 10 μm or 5 μm. **b**–**f** 24 h after the last OVA challenge, mice were subjected to examination of methacholine-provoked airway hyperreactivity (**f**, n = 6, P values: 0.0002, 0.0003, <0.0001, <0.0001, <0.0001, <0.0001; 0.0221, 0.0046, 0.0018, 0.0348, 0.025; 0.0029, 0.0002, 0.0001, 0.0004, <0.0001). BALFs were prepared for cell counting and classification (**c**, n = 6, P values: <0.0001, <0.0001, <0.0001, <0.0001, <0.0001; 0.0112, 0.0424, 0.0098; <0.0001, 0.0155, <0.0001), and lungs were harvested for paraffin-embedded sectioning for H&E staining (**b**, scale bar, 10 μm or 5 μm), PAS staining and immunostaining of Muc5AC (**d**, scale bar, 10 μm), their semi-quantification. (**e**, n = 6, P values: <0.0001, <0.0001, <0.0001; <0.0001, <0.0001, <0.0001). **g** Mice were intratracheally instilled with IL-13 (12 μg/mouse) to induce airway goblet cell metaplasia, following intratracheal instillation of 2-D08 at the dosages of 10 and 30 mg/kg. **h**, **i** Lungs were subjected to paraffin-embedded sectioning for PAS and Muc5AC staining (**h**, scale bar, 10 μm) and their semi-quantification (**i**, n = 6, P values: <0.0001, <0.0001, <0.0001; <0.0001, <0.0001, <0.0001). Mean ± SD, One-way ANOVA and Tukey-Kramer multiple comparisons test, *, +P < 0.05, **, ++P < 0.01. Source data are provided as a Source Data file.

intratracheally instilled on day 9 to inhibit the protein SUMOylation, and mice were then sampled on day 16 for the following analyses (Fig. 4a). Knock-in of *caRhoA* in airway epithelia or/and intratracheal instillation of 2-D08 affected neither the number and classification of immune cells in BALFs and nor the apparent infiltration of immune cells into peribronchial and perivascular tissues (Fig. 4b, c). Notably, knock-in of *caRhoA* in airway epithelia robustly increased the PAS, Muc5AC, and p-ROCK2 staining in the bronchial epithelia, whereas intratracheal instillation of 2-D08 at 10 or 30 mg/kg markedly attenuated or completely abolished these effects of *caRhoA*, respectively (Fig. 4c–f). In addition, western analyses of bronchial p-ROCK2 levels consistently indicated 2-D08 at 10 and 30 mg/kg dose-dependently decreased the

caRhoA-induced p-ROCK2 levels (Fig. 4g, h). Thus, activation of RhoA alone was sufficient to induce the airway epithelial metaplasia, and suppression of SUMOylation consistently abolished RhoA-induced ROCK2 activation and subsequent airway epithelial metaplasia.

**SUMOylation on K1007 activates ROCK2**

To investigate further the role of SUMOylation in activating ROCK2, we transfected 293 T cells with Myc-tagged SUMO1. SUMO1 increased the p-ROCK2 and p-MLC2 in both the dose- and time-dependent manners (Fig. 5a, b). Co-immunoprecipitation experiments indicated protein complexes precipitated by a ROCK2 antibody contained abundant SUMOylated ROCK2 in 293 T cells, 16HBE cells and

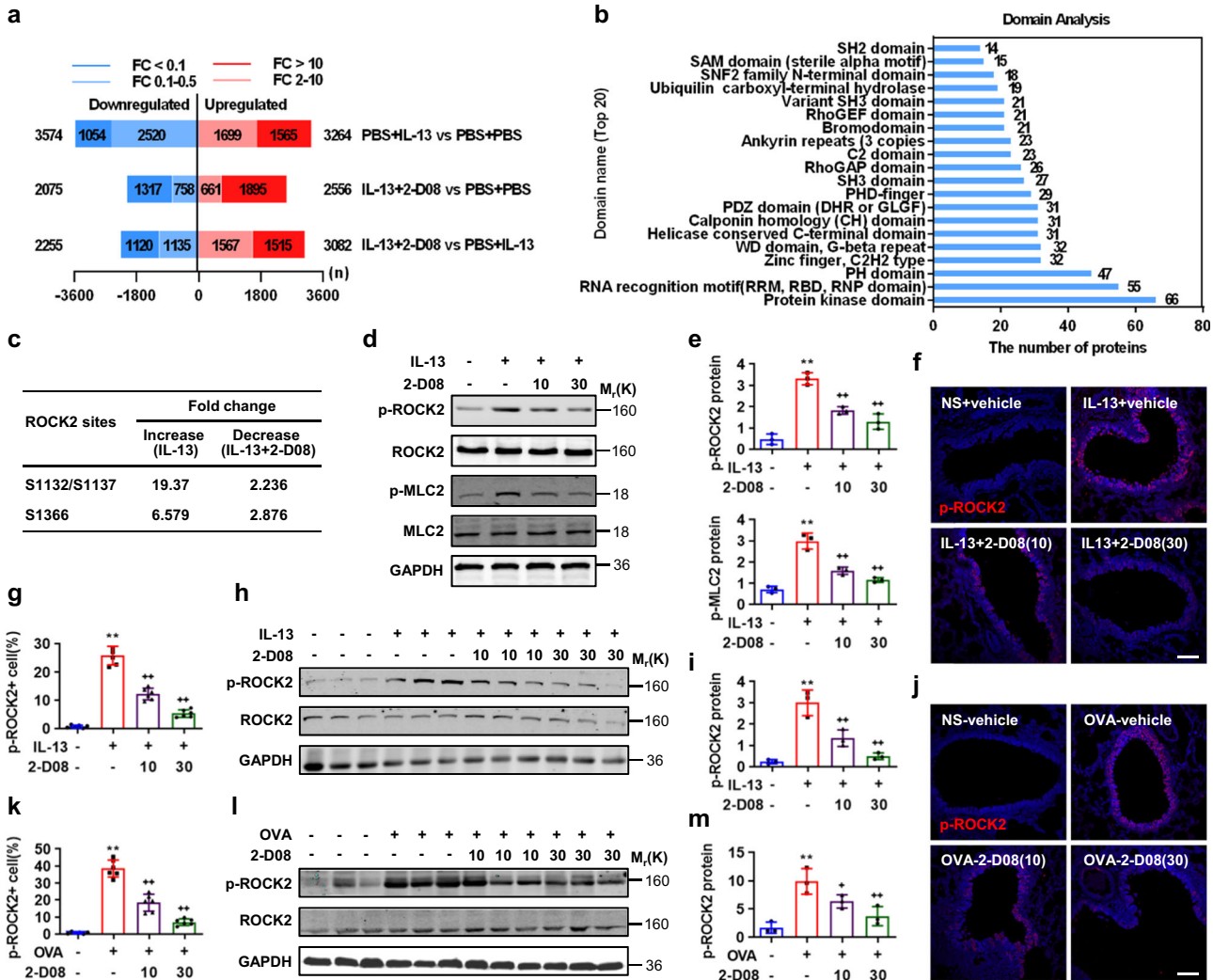

**Fig. 3 | Suppression of SUMOylation inactivates ROCK2 in response to OVA or IL-13 stimulation in bronchial epithelia or 16HBE cells. a** Quantitative phosphoproteomic analysis in 16HBE cells treated with IL-13 (100 ng/ml) at 6 h post and during pretreatment with 2-D08 (30 µM). The bar graph shows the quantitative results of phosphorylated sites. **b** Enrichment and analysis of the protein domains of differentially expressed phosphorylated peptides. **c** The altered phosphorylation levels on the key phosphorylation sites of ROCK2 upon IL-13 or/and 2-D08 treatment. **d**, **e** Western analyses in 16HBE cells treated with IL-13 at 6 h post and during

treatment with 2-D08 (n = 3, P values: 0.0002, 0.0015, 0.0016; <0.0001, 0.0076, 0.0003). **f**–**m** Lungs and bronchi from Fig. 2g and Fig. 2a were subjected to immunofluorescence staining and western analyses, respectively, each n = 6 or 3, P values: <0.0001, <0.0001, <0.0001 (**g**); 0.0015, 0.009, 0.0023 (**i**); <0.0001, <0.0001, <0.0001 (**k**); 0.0022, 0.0364, 0.0089 (**m**). Scale bar, 5 µm. Mean ± SD, One-way ANOVA and Tukey-Kramer multiple comparisons test, ⁺P < 0.05, ⁺⁺, **P < 0.01. Source data are provided as a Source Data file.

mouse primary bronchial epithelial cells (Fig. 5c and Supplementary Fig. 4a), suggesting the potential SUMOylation of ROCK2. Likewise, protein complexes precipitated by a Flag antibody contained a larger amount of SUMOylated Flag-ROCK2 in 293T cells and 16HBE cells expressing Myc-SUMO1, Flag-ROCK2, and scramble shRNA than in those expressing Myc-SUMO1, Flag-ROCK2, and UBC9-shRNA (Fig. 5d and Supplementary Fig. 4b). To determine whether ROCK2 was indeed SUMOylated, we performed in vitro SUMOylation assay. SUMOylation of ROCK2 protein was readily detectable in the reaction mixture containing recombinant ROCK2 protein and incubated with E1, E2 and SUMO1 at 37 °C instead of at 4 °C or without E1, E2 or SUMO1 (Fig. 5e).

To identify the potential SUMOylation site(s) on ROCK2, we performed bioinformatic analysis and found several conserved SUMOylation consensus motifs including K88, K216, K238, K355, K615, K795, K828, K884, K1007, K1049, and K1071 in human ROCK2 sequence (Supplementary Fig. 5). To assess their potential importance, we expressed ROCK2 variants that harbor mutations at the consensus lysine (K) residues (Lys to Arg) individually and evaluated their impact

on phosphorylation of MLC2. Wild-type (WT) ROCK2 and these variants were comparably expressed in the 293T cells. Notably, ROCK2(K1007R) mutant caused a robust decrease in p-MLC2 as compared to ROCK2(WT) (Fig. 5f), suggesting that K1007R mutation inactivated ROCK2 and SUMOylation and deSUMOylation on K1007 could affect the kinase activity of ROCK2. This hypothesis was confirmed by the co-immunoprecipitation and western analyses which indicated immunocomplex precipitated by a Flag antibody contained abundant Myc-SUMO1-conjugated ROCK2(WT) but relatively less Myc-SUMO1-conjugated ROCK2(K1007R) in 293T cells and 16HBE cells expressing Myc-SUMO1 and Flag-ROCK2 variants (Fig. 5g and Supplementary Fig. 4c), and SUMO1 or UBC9-shRNA increased or decreased the phosphorylation levels of Flag-ROCK2(WT) but not of Flag-ROCK2(K1007R) in 293T cells, respectively (Fig. 5h). Analysis of in vitro SUMOylation consistently indicated ROCK2(WT) or ROCK2(K1007R) was largely or rarely SUMOylated in the reaction mixture containing E1, E2 and SUMO1 at 37 °C, respectively (Fig. 5i). Thus, ROCK2 was SUMOylated or deSUMOylated on K1007 to be activated and inactivated, respectively.

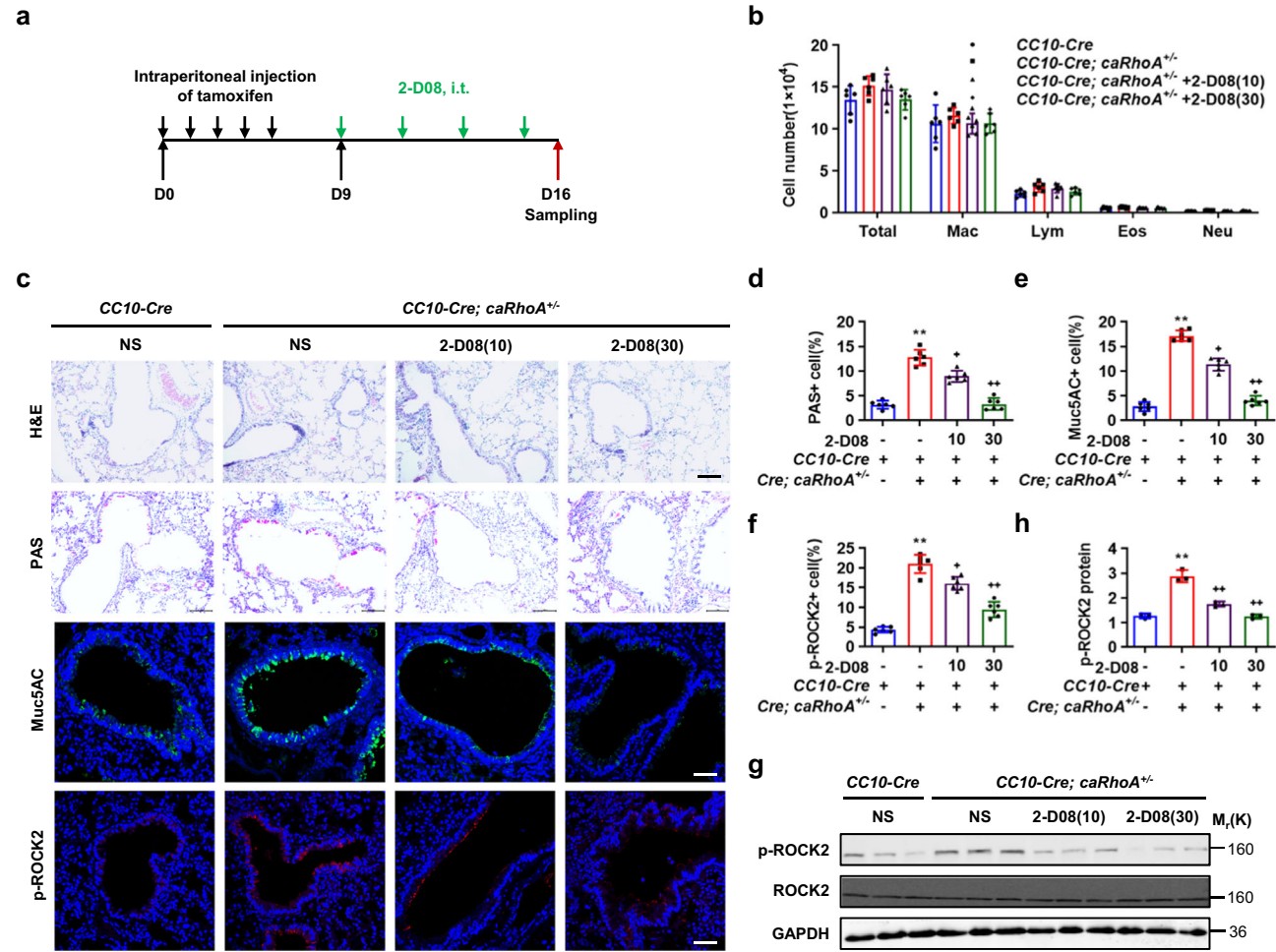

**Fig. 4 | Suppression of SUMOylation attenuates ROCK2 activation and airway goblet cell metaplasia in response to RhoA activation. a** *CC10-Cre* and *CC10-Cre; caRhoA*[+/-] mice at 8 weeks of age were peritoneally injected with tamoxifen at 200 mg/kg on day 0, 1, 2, 3, and 4, and then intratracheally received 2-D08 at 10 or 30 mg/kg on day 9, 11, 13, and 15. Mice were euthanized on day 16 for the following analyses. **b** Cell counting and classification in BALFs. **c**–**f** H&E and PAS staining

(scale bar, 10 μm), and immunostaining for Muc5AC and p-ROCK2 (scale bar, 5μm) in lung sections and their semi-quantification. *P* values: <0.0001, 0.0107, <0.0001 (**d**); <0.0001, 0.0134, <0.0001 (**e**); <0.0001, 0.0135, <0.0001 (**f**). **g, h** Western analyses and semi-quantification for bronchi (*P* values: 0.0008, 0.0019, 0.0004). Mean ± SD, *n* = 6, One-way ANOVA and Tukey-Kramer multiple comparisons test, *· +*P* < 0.05, **· ++*P* < 0.01. Source data are provided as a Source Data file.

To determine the role of SUMOylation on K1007 in activating ROCK2 in response to IL-13 and RhoA, we performed western analyses. Flag-ROCK2(K1007R) decreased not only the basal but also the IL-13- or caRhoA-induced p-ROCK2 phosphorylation levels, as compared to Flag-ROCK2(WT) (Fig. 5j, k). K1007 consensus motif locates at the Rho-binding domain of ROCK2[26], we next performed co-immunoprecipitation and GST pull-down experiments to explore the potential mechanism underlying K1007R mutation inactivating ROCK2. Protein complexes precipitated with a Flag antibody and blotted with a HA antibody contained much more HA-RhoA in 16HBE cells expressing Flag-ROCK2(WT) than in those expressing but little Flag-ROCK2(K1007R), especially in the presence of IL-13 stimulation (Fig. 5l). Moreover, GST pull-down assays for active form of RhoA (GTP-RhoA) indicated GTP-RhoA interacted with significantly more Flag-ROCK2(WT) than Flag-ROCK2(K1007R) and GTP-RhoA consistently interacted with much more p-ROCK2 in 16HBE cells expressing Flag-ROCK2(WT) than in those expressing Flag-ROCK2(K1007R) (Fig. 5m). Finally, immunofluorescence staining for RhoA and p-ROCK2 in bronchial epithelia of OVA-sensitized and -challenged mice demonstrated that OVA challenge significantly enhanced the overlapping immunosignals derived from RhoA and p-ROCK2, whereas intratracheal 2-D08 administration

robustly reduced the OVA-induced co-localization of RhoA and p-ROCK2 in bronchial epithelia (Supplementary Fig. 6a, b). Taken together, SUMOylation of ROCK2 on K1007 activated ROCK2 by facilitating its binding to and phosphorylation by RhoA.

## PIAS1 functions as an E3 ligase to SUMOylate ROCK2 on K1007

SUMO E3 ligases play a key role in specifying protein substrates for SUMOylation. PIAS family of E3 ligases consists of four PIASs, including PIAS1, PIASx (PIAS2), PIAS3, and PIASy (PIAS4), and specify a diverse set of proteins for SUMOylation[27-29]. We then investigated the potential roles of PIASs in SUMOylation and activation of ROCK2. Overexpression of PIAS1 instead of PIAS2, 3 and 4 robustly induced the p-ROCK2 levels. Conversely, PIAS1 but not PIAS2, 3, and 4 siRNA significantly reduced SUMO1- or PIAS1-induced p-ROCK2 levels (Fig. 6a–c and Supplementary Fig. 7a, b). The physical interaction between PIAS1 and ROCK2 was confirmed by co-immunoprecipitation experiments which indicated immunocomplex precipitated by a Flag antibody contained a large amount of Myc-PIAS1 in 293 T cells expressing Myc-PIAS1 and/or Flag-ROCK2 (Fig. 6d). To determine whether PIAS1 SUMOylates ROCK2 on K1007, we performed western and co-immunoprecipitation experiments. Overexpression of Myc-PIAS1 markedly increased the p-ROCK2 levels in Flag-ROCK2(WT)

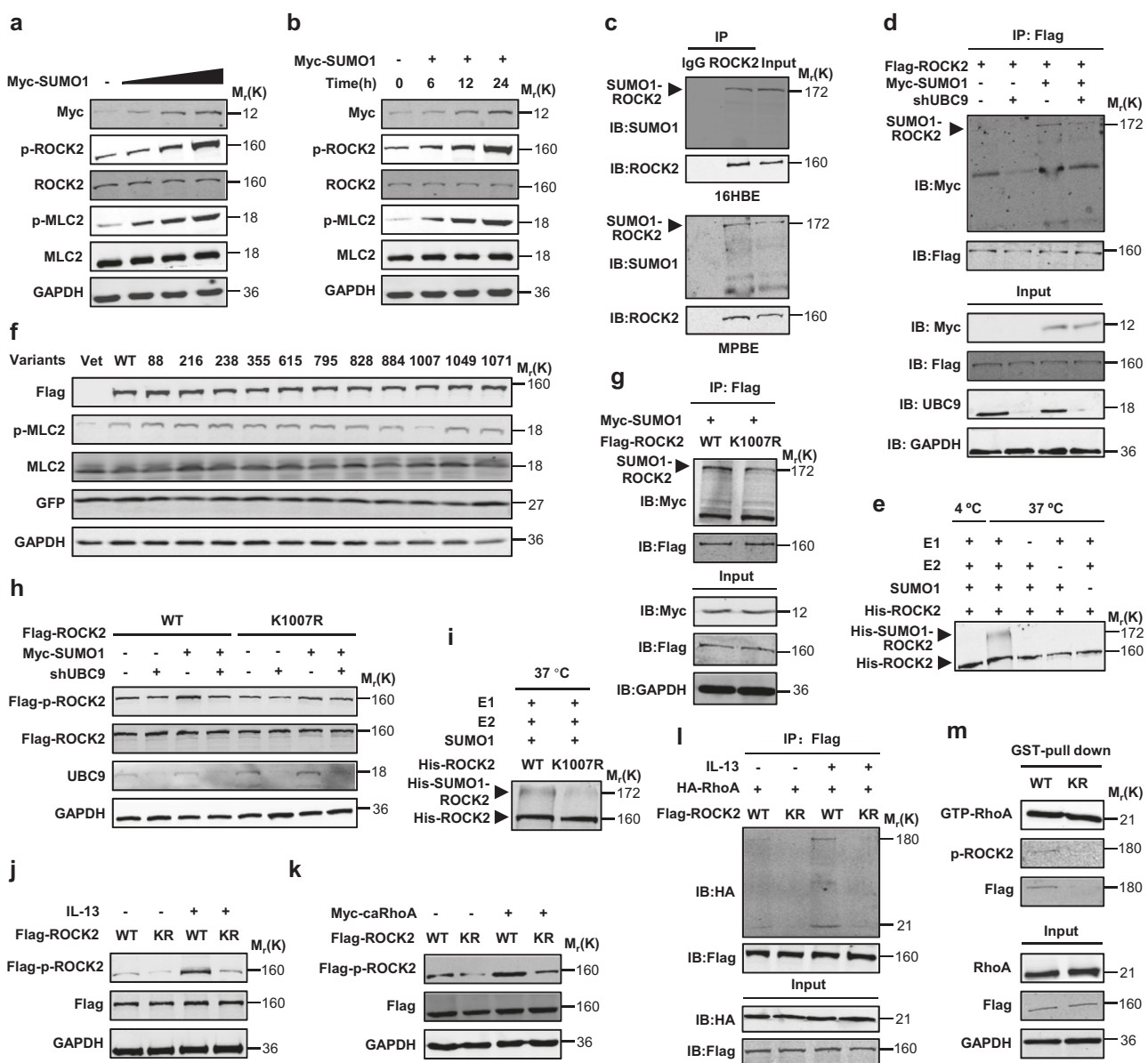

**Fig. 5 | ROCK2 is SUMOylated on K1007 to be activated. a, b** Western analyses in 293T cells transfected with Myc-SUMO1 for 24 h or the indicated times. **c** Co-immunoprecipitation experiments using a control IgG or a ROCK2 antibody in 16HBE cells and mouse primary bronchial epithelial cells (MPBEs). **d** Co-immunoprecipitation experiments using a Flag antibody in 16HBE cells transfected with Flag-ROCK2 in combination with Myc-SUMO1 in the presence of scramble or UBC9 shRNA. **e** In vitro SUMOylation assays in a reaction mixture containing ROCK2 recombinant protein, E1, E2, and SUMO1 and incubated at 37 °C or 4 °C for 60 min, followed by western analyses. **f** Western analyses in 293T cells 24 h after transfection with vector, wild-type (WT) ROCK2 or ROCK2 variants (K → R). **g** Co-immunoprecipitation experiments using a Flag antibody in 16HBE cells transfected with Myc-SUMO1 and Flag-ROCK2/Flag-ROCK2(K1007R). **h** Western analyses in

293T cells at 24 h post-transfection with or without Myc-SUMO1, UBC9 shRNA, and Flag-ROCK2 /Flag-ROCK2(K1007R). **i** In vitro SUMOylation assays in a reaction mixture containing ROCK2(WT) or ROCK2(K1007R) recombinant protein, E1, E2, and SUMO1, followed by western analyses. **j, k** Western analyses in 293 T cells transfected with Flag-ROCK2(WT or K1007R) in the presence of IL-13 stimulation for 6 h or of Myc-caRhoA. **l** Co-immunoprecipitation experiments using a Flag antibody in 16HBE cells transfected with HA-RhoA and Flag-ROCK2(WT or K1007R) after IL-13 treatment for 6 h. **m** 16HBE cells transfected with Flag-ROCK2(WT or K1007R), were subjected to GST pull-down assays with Rhotekin-RBD-coated beads, followed by western analyses. Experiments were repeated independently at least three times with similar results. Source data are provided as a Source Data file.

but not in Flag-ROCK2(K1007R) (Fig. 6e), and immunocomplex precipitated by a Flag antibody contained significantly more Myc-PIAS1 in 293T cells expressing Flag-ROCK2(WT) than in those expressing Flag-ROCK2(K1007R) (Fig. 6f). Thus, PIAS1 functioned as an E3 ligase to participate in the SUMOylation on K1007 and activation of ROCK2.

To explore the expression pattern of PIAS1 in healthy and asthmatic human bronchi, we performed immunostaining. PIAS1 was specifically and apparently detected in the epithelia of human lobar bronchi (Fig. 6g), and was significantly more localized in BALF CC10⁺

cells of patients with allergic asthma than with FBA (Fig. 6h, i). In addition, PIAS1 was more robustly expressed in the bronchial epithelia and bronchi of OVA- than of NS-challenged mice (Fig. 6j–m). To determine whether PIAS1 was involved in the allergic airway goblet cell metaplasia, we intratracheally instilled lentiviral PIAS1-shRNA and IL-13 into mice and performed H&E, PAS, and immunofluorescence staining. PIAS1-shRNA significantly decreased IL-13-induced PAS, Muc5AC and p-ROCK2 staining in the bronchial epithelia (Fig. 6n, o and Supplementary Fig. 7c, d), and consistently decreased the protein expression of IL-13-induced p-ROCK2 and p-MLC2 levels in bronchi (Fig. 6p, q).

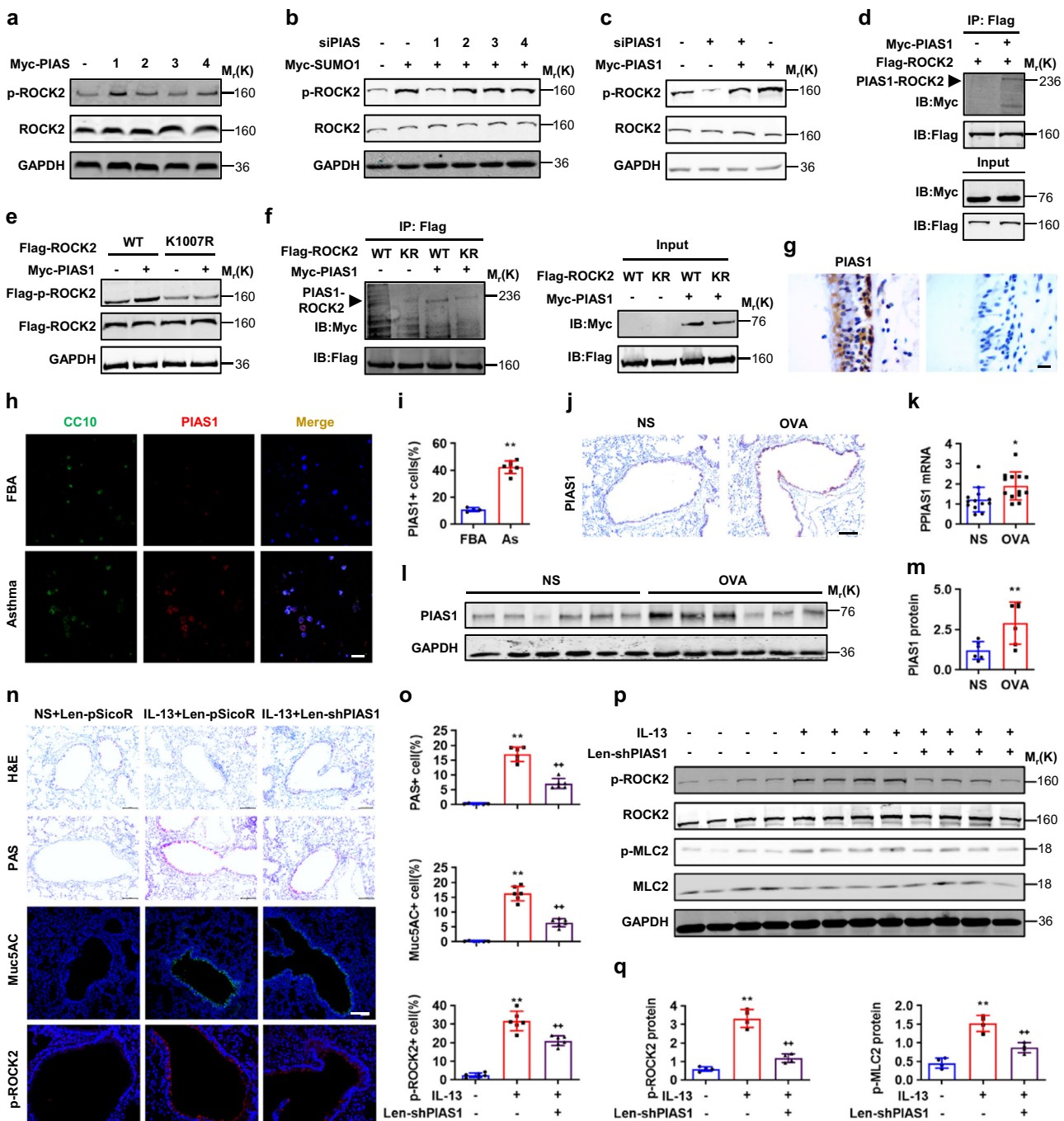

**Fig. 6 | SUMOylation activates ROCK2 through PIAS1 E3 ligases. a, b** Western analyses in 293T cells at 48 h post transfection with Myc-PIAS1, 2, 3, 4 or siRNAs of PIAS1, 2, 3, 4 in combination with or without Myc-SUMO1. **c** Western analyses in 293T cells at 48 h post transfection with or without Myc-PIAS1 and PIAS1 siRNA. **d** Co-immunoprecipitation experiments in 293T cells at 48 h post transfection with or without Myc-PIAS1 and Flag-ROCK2. **e, f** Western and co-immunoprecipitation analyses in 293T cells at 48 h post transfection with or without Myc-PIAS1 and Flag-ROCK2(WT or K1007R). **g** IHC of PIAS1 in human healthy bronchial sections (n = 4, left: experimental group, right: negative control group, scale bar, 2 μm). **h, i** Immunostaining of CC10, PIAS1, DAPI and semi-quantification in BALF cells from children with FBA or asthma (P value: <0.0001, scale bar, 10 μm). **j–m** Lungs or bronchi from NS- and OVA-challenged mice were subjected to IHC (**j**), qPCR (**k**, P value: 0.0127), and western analyses (**l**) and their semi-quantification (**m**, P value: 0.0093). Scale bar, 10 μm. **n–q** Mice were intratracheally instilled with lentiviral scramble- or PIAS1-shRNA and then with IL-13. Lungs and bronchi were subjected to H&E (scale bar, 10 μm), PAS (scale bar, 10 μm), Muc5AC (scale bar, 10 μm) and p-ROCK2 (scale bar, 5 μm) staining (**n, o**, P values: <0.0001, <0.0001; <0.0001, <0.0001; <0.0001, 0.0014) and western analyses (**p, q**, P values: <0.0001, 0.0002; 0.0002, 0.0031). Mean ± SD, n = 4, unpaired two-tailed Student's t test or One-way ANOVA and Tukey-Kramer multiple comparisons test, *, +P < 0.05, **, ++P < 0.01. Experiments were repeated independently at least three times with similar results. Source data are provided as a Source Data file.

Thus, high expression of PIAS1 in bronchial epithelia of the patients and mouse model with allergic asthma promoted the SUMOylation and activation of ROCK2 and thereby induced the airway goblet cell metaplasia.

## DeSUMOylation of ROCK2 on K1007 attenuates airway goblet cell metaplasia in response to allergen or IL-13 stimulation

To investigate further the role of SUMOylation-mediated ROCK2 activation in allergic airway goblet cell metaplasia, we constructed

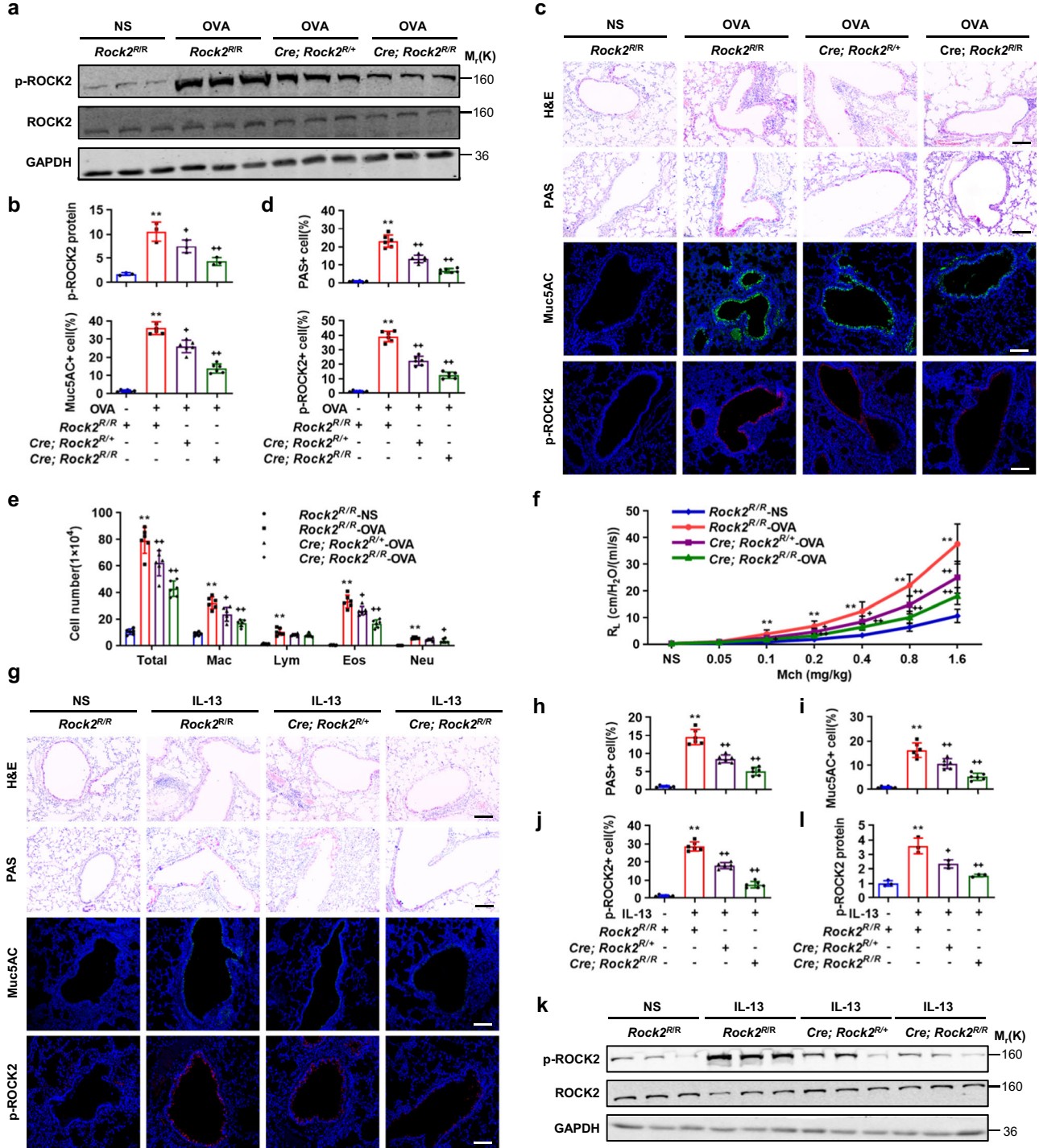

**Fig. 7 | DeSUMOylation of ROCK2 on K1007 attenuates airway goblet cell metaplasia in response to allergen or IL-13 stimulation. a**, **b** *Rock2^{K1007R/K1007R}*, *CC10-Cre; Rock2^{K1007R/+}* and *CC10-Cre; Rock2^{K1007R/K1007R}* mice were sensitized and challenged with OVA as described in Fig. 2a, and bronchi were then subjected to western analyses and semi-quantification (each *n* = 3, *P* values: 0.0016, 0.0486, 0.0073). **c**, **d** Lungs (each *n* = 6) were subjected to paraffin-embedded sectioning, H&E and PAS staining, immunostaining for Muc5AC and p-ROCK2 (**c**, scale bar, 10 μm), and their semi-quantification (**d**, *P* values: <0.0001, 0.0001, <0.0001; <0.0001, 0.0115, <0.0001; <0.0001, <0.0001, <0.0001). **e**, **f** BALF cell counting and classification (**e**, *P* values: <0.0001, <0.0001, <0.0001, <0.0001, <0.0001; 0.0006, 0.0437, 0.0159; <0.0001, <0.0001, 0.0006, 0.0155) and methacholine-provoked

airway hyperreactivity (**f**, each *n* = 6, *P* values: 0.0002, 0.0001, <0.0001, <0.0001, <0.0001; 0.049, 0.0399, 0.0073, 0.0095; 0.0196, 0.0019, 0.0035, 0.0002, 0.0003). **g**–**l** *Rock2^{K1007R/K1007R}*, *CC10-Cre; Rock2^{K1007R/+}* and *CC10-Cre; Rock2^{K1007R/K1007R}* mice were intratracheally instilled with or without IL-13 as described in Fig. 2g, and lungs or bronchi were subjected to H&E and PAS staining and immunostaining for Muc5AC and p-ROCK2 (**g**–**j**, each *n* = 6, *P* values: <0.0001, 0.0001, <0.0001; <0.0001, 0.0036, <0.0001; <0.0001, <0.0001, <0.0001) or western analyses (**k**, **l**, each n = 3, *P* values: 0.0015, 0.0236, 0.0034). Scale bar, 10 μm. Mean ± SD, One-way ANOVA and Tukey-Kramer multiple comparisons test, *, ⁺*P* < 0.05, **, ⁺⁺*P* < 0.01. Source data are provided as a Source Data file.

 

conditional ROCK2(K1007R) knock-in mice and crossed them with *CC10-Cre^ERT2* mice to generate *CC10-Cre^ERT2; Rock2^K1007R/+* and *CC10-Cre^ERT2; Rock2^K1007R/K1007R* mice and their age- and sex-matched control littermates (*Rock2^K1007R/K1007R*). Mice at 8 weeks of age were subjected to tamoxifen induction and OVA challenge. *CC10-Cre^ERT2; Rock2^K1007R/+* and *CC10-Cre^ERT2; Rock2^K1007R/K1007R* mice did not show any apparent airway epithelial goblet cell metaplasia and inflammatory cells infiltration into peribrochial and perivascular tissues, as compared to *Rock2^K1007R/K1007R* mice after tamoxifen induction and NS challeng. Notably, western analyses indicated *CC10-Cre^ERT2; Rock2^K1007R/+* and *CC10-Cre^ERT2; Rock2^K1007R/K1007R* mice exhibited a dose-dependent decrease in p-ROCK2 levels in the bronchi, as compared to *ROCK2^K1007R/K1007R* mice, in response to OVA challenge (Fig. 7a, b). H&E, PAS, Muc5AC, and p-ROCK2 staining demonstrated that *CC10-Cre^ERT2; Rock2^K1007R/+* and *CC10-Cre^ERT2; Rock2^K1007R/K1007R* mice dose-dependently suppressed the OVA-induced the infiltration of inflammatory cells into peribronchial and perivascular tissues and the staining of PAS, Muc5AC, and p-ROCK2 in bronchial epithelia, as compared to *Rock2^K1007R/K1007R* mice (Fig. 7c, d). Correspondingly, cell counting and classification of BALF cells revealed that *CC10-Cre^ERT2; Rock2^K1007R/+* and *CC10-Cre^ERT2; Rock2^K1007R/K1007R* mice decreased OVA-induced numbers of total inflammatory cells, macrophages, eosinophils, and neutrophils in a dose-dependent manner, as compared to *Rock2^K1007R/K1007R* mice (Fig. 7e). Likewise, spirometry results indicated that *CC10-Cre^ERT2; Rock2^K1007R/+* and *CC10-Cre^ERT2; Rock2^K1007R/K1007R* mice significantly decreased airway resistance induced by methacholine ranging from 0.2 to 1.6 mg/kg and from 0.1 to 1.6 mg/kg, respectively (Fig. 7f). Thus, deSUMOylation of ROCK2 on K1007 inactivated ROCK2 to attenuate airway goblet cell metaplasia in response to allergen stimulation.

To determine whether deSUMOylation of ROCK2 on K1007 directly suppresses allergic airway goblet cell metaplasia, we next intratracheally instilled IL-13 into *Rock2^K1007R/K1007R*, *CC10-Cre^ERT2; Rock2^K1007R/+* and *CC10-Cre^ERT2; Rock2^K1007R/K1007R* mice, following tamoxifen induction for 4 days, and performed western analyses and histological assessments. As expected, *CC10-Cre^ERT2; Rock2^K1007R/+* or *CC10-Cre^ERT2; Rock2^K1007R/K1007R* mice largely decreased or completely abolished the IL-13-induced PAS, Muc5AC, and p-ROCK2 staining in bronchial epithelia, as compared to *Rock2^K1007R/K1007R* mice, respectively (Fig. 7g–j). Likewise, western analyses indicated *CC10-Cre^ERT2; Rock2^K1007R/+* or *CC10-Cre^ERT2; Rock2^K1007R/K1007R* mice mildly or robustly decreased the IL-13-induced bronchial p-ROCK2 levels, as compared to *Rock2^K1007R/K1007R* mice, respectively (Fig. 7k, l). Thus, deSUMOylation of ROCK2 on K1007 in bronchial epithelia inactivated ROCK2 to attenuate airway goblet cell metaplasia in response to IL-13 stimulation.

## Discussion

By biochemical and genetic approaches, we have uncovered the protein SUMOylation in the regulation of goblet cell metaplasia in allergic airway epithelia and identified the ROCK2 SUMOylation on K1007 and subsequent its activation as a hitherto uncharacterized mechanism controlling goblet cell metaplasia and mucus hypersecretion that contribute to airway hyperreactivity in allergic asthma. In this molecular event, SUMOylation of ROCK2 on K1007 enhances ROCK2's binding to and activation by RhoA. Thus, SUMOylation-mediated ROCK2 activation is an integral component of RhoA/ROCK signaling and is an additional mechanism governing this signaling pathway.

The present study is consistent with previous findings indicating that enrichment of the SUMOylation pathway represents a potential key regulator of airway remodeling[23], and that SUMOylation is enhanced in the airway epithelia and inhibition of SUMOylation attenuates Th2 cell inflammation in allergen-induced mouse model of asthma, where chromobox 4 (CBX4), an E3 ligase, participates in the airway inflammation[30,31]. However, the present study has further identified that ROCK2 is a substrate of SUMOylation and K1007 is a

determinant site for SUMOylation-mediated ROCK2 activation and subsequent allergic airway goblet cell metaplasia.

In OVA- or HDM-induced mouse model of asthma, intratracheal instillation of 2-D08 attenuates not only airway inflammation but also airway goblet cell metaplasia. We assume that the attenuation of airway goblet cell metaplasia by 2-D08 is resulted from its direct inhibition of SUMOylation in airway epithelia and its secondary effect on attenuation of airway inflammation as well. This hypothesis is supported by the following findings: (1) 2-D08 is sufficient to alleviate airway epithelial goblet cell metaplasia in response to either intratracheal IL-13 instillation or airway epithelial activation of RhoA, either of which doesn't apparently evoke the airway inflammation; (2) Intratracheal knockdown of PIAS1 consistently decreases IL-13-induced airway goblet cell metaplasia but has no effect on airway inflammation; (3) Airway epithelial knock-in ROCK2(K1007R) robustly decreases both the allergen- and IL-13-induced airway goblet cell metaplasia.

RhoA activation is sufficient to induce airway epithelial goblet cell metaplasia, though airway epithelial knock-in ROCK2(K1007R) affects neither the airway goblet cell metaplasia nor the airway inflammation in intact mice. Interestingly, airway epithelial knock-in ROCK2(K1007R) markedly decreases not only airway goblet cell metaplasia but also airway inflammation in response to allergen stimulation. This could be due primarily to the ROCK2 inactivation resultant decrease in inflammatory mediators secreted from airway epithelia. This notion is supported by the following previous findings that ROCK2 is implicated in regulation of interleukin-21 (IL-21) and IL-17 secretion in mice and humans[32,33], that cigarette smoking inhibits ROCK2 activation to modulate IL-22 production in T cells[34], that Benzo(a)pyrene (BaP) co-exposure with dermatophagoides group 1 allergen (Derf1) induces transforming growth factor-β1 (TGF-β1) secretion through RhoA/ROCK signaling in human bronchial epithelial cells and airways of asthma mouse model[16], that loss of p120 increases RhoA and NF-κB activity and thereby induces IL-8 expression in bronchial epithelial cells[35], and that ROCK-dependent signaling pathway regulates inducible nitric oxide synthase (NOS) expression and NO production in airway epithelial cells[36,37].

Importantly, we have identified that PIAS1 as a SUMO E3 ligase for ROCK2 SUMOylation on K1007 and activation. This observation is consistent with a previous quantitative SUMO proteomic study that identifies PIAS1 substrates involved in cell migration and motility[38], with an our previous study indicating that large tumor suppressor 1, a serine/threonine kinase, is SUMOylated by PIAS1 on K830[39], and with previous other studies showing that PIAS1 SUMOylates focal adhesion kinase (FAK), c-Jun N-terminal kinase (JNK), RhoB and consequently regulates their kinase activities[40–42]. Though airway epithelial knockdown of PIAS1 is sufficient to attenuate IL-13-induced airway goblet cell metaplasia, its potential role in allergen-induced mouse model of asthma remains unknown. Since targeting the PIAS1 SUMO ligase pathway is capable of controlling inflammation and PIAS1 has recently been identified as a component of the transcription-coupled repair complex other than a SUMO E3 ligase[43,44], further experiments are thus needed to clarify its exact role in the allergic airway diseases.

Our quantitative phosphoproteomics data show that 625 IL-13-induced phosphorylation sites were significantly downregulated after 2-D08 treatment in 16HBE cells. This implicates that an abundant phosphoproteins are SUMOylated in the allergic airway epithelia and ROCK2 is only a representative of these phosphoproteins. Quantitative SUMO proteomics experiments are thus needed to identify more SUMOylation substrates in the allergic airway epithelia. Importantly, deSUMOylation mutation of ROCK2 on K1007R almost completely abolishes the kinase activity of ROCK2, and ROCK2 but not ROCK1 participates in the goblet cell hyperplasia in mouse model of allergic asthma[17]. Given the fact that Rho/ROCK signaling is tightly involved in the airway inflammation, hyperreactivity, and epithelial remodeling[45,46], ROCK2 SUMOylation signaling pathway could plays a

 

determinant role in the cardinal features of allergic airway diseases, including but not limited to the goblet cell metaplasia.

Taken together, we have identified SUMOylation-mediated ROCK2 activation is an integral component of Rho/ROCK signaling pathway and plays a determinant role in controlling the airway goblet cell metaplasia and most likely the overall features of allergic airway diseases. Blockage of this SUMOylation pathway is thus a promising or an additional approach for therapeutic intervention of allergic asthma.

## Methods

### Inclusion and ethics

Ethical permissions were obtained for all studies including human-derived material presented and mice in this study. The usage of human healthy lobar bronchial tissues in the present study was approved by the Clinical Research Ethics Committee of the Affiliated Hospital of Jiaxing University, permission number LS2020-171. Written consents for the usage of bronchial tissues in the present study were obtained with no financial incentive provided. The study using the remnant BALFs from children with a diagnosis of FBA or allergic asthma was approved by the Ethics Committee of the Children's Hospital of Zhejiang University School of Medicine, permission number 2015-HP-037. Written consent for diagnostic fiberoptic bronchoscopy with BAL and written consent for the use in research of remnant BALFs were obtained from the children's guardians. Mice experimental protocols were approved by the Zhejiang University Institutional Animal Care and Use Committee, permission number ZJU202017549.

### Clinical specimens

Human healthy lobar bronchi were surgically obtained from patients, who had peripheral lung cancers and underwent pulmonary lobectomy at Department of Thoracic Surgery, the Affiliated Hospital of Jiaxing University (Jiaxing, China). Lobar bronchial tissues with sizes of ~2 cm³ were surgically removed as far as possible from tumor location and fixed for histological examination and immunofluorescence staining.

### BALFs from children with FBA or allergic asthma

BALFs from children with a diagnosis of FBA or allergic asthma were obtained as described previously[47,48]. Children with the suspected diagnoses of FBA, tuberculosis, or bronchomalacia received diagnostic fiberoptic bronchoscopy with bronchoalveolar lavage at the Zhejiang University School of Medicine Children's Hospital. The remnant BALFs from children with discharged diagnoses of FBA or allergic asthma were used in the current study. BALFs were subjected to preparation of cytospins and supernatants and the cytospins of BALFs from children with either FBA or asthma were further prepared for the immunofluorescence staining of SAE1, SAE2, UBC9 and CC10.

### Mouse strains

C57BL/6J and BALB/c mice at 8 weeks of age were purchased from SLAC Laboratory Animal Co. Ltd. (Shanghai, China). The *CC10-Cre*[ERT2] mouse strain with genetic background of C57BL/6J was purchased from Model Animal Research Center of Nanjing University (Nanjing, China). The conditional *caRhoA* (*caRhoA*[+/−]) knock-in mouse strain with genetic background of C57BL/6J was generated by Cyagen Biosciences (Santa Clara, CA) as described previously[13], and the conditional *Rock2*[K1007R/+] knock-in founders with genetic background of C57BL/6 J were generated by CRISPR/Cas9 at Cyagen Biosciences as described previously[49]. All animals were housed in a room maintained at 23 °C ± 2 °C with 50% ± 10% humidity and a 12-h light/12-h dark cycle, fed with standard chow from Xietong Pharmaceutical Bio-engineering Co. Ltd. (SFS9112)

and free access to water under specific pathogen-free (SPF) conditions at the Zhejiang University Animal Care Facility according to the Institutional Guidelines for Laboratory Animals.

### A mouse model with OVA-induced allergic airway inflammation

Female BALB/c mice at 8 weeks of age were used due to their higher susceptibility to allergic airway inflammation[50]. An OVA-induced mouse model with allergic airway inflammation was prepared as described previously[48,51]. Briefly, on day 0, all mice were subcutaneously injected with 20 μg/mouse of OVA (Sigma, St. Louis, MO) emulsified in 2 mg aluminum hydroxide adjuvant at the footpad, neck, back, and groin. On day 14, mice were further intraperitoneally injected with 0.2 ml/mouse of 0.2% OVA. From day 22 to 28, the sensitized mice were atomized with 1% OVA or NS by a jet nebulizer (BARI Co. Ltd., Germany) for 30 min, once daily, and 2 h after each OVA atomization, mice were intratracheally administered with either 50 μl/mouse of 2-D08 (Sigma) at the dosages of 10 and 30 mg/kg or vehicle. On day 29, mice were subjected to the examination of methacholine-provoked airway hyperreactivity and were then euthanized by intraperitoneal injection of excess pentobarbital sodium for sampling, and the preparation of BALFs.

### A mouse model with HDM-induced allergic airway inflammation

The HDM-induced mouse model with allergic airway inflammation was prepared as described previously[52,53]. Briefly, Mice at 8 weeks of age were sensitized by intranasal administration with 25 μg of HDM extract (Greer Laboratories Inc, Lenoir, NC) in 20 μl of NS (1.25 mg/ml) or equal volume of NS, five days a week for three weeks. From day 21 to 27, mice were atomized with 0.5 ‰ HDM to trigger recall responses or an equal volume of NS for 30 min, once daily, and 50 μl/mouse of 2-D08 (Sigma) at the dosages of 10 and 30 mg/kg or vehicle was intratracheally administered at 2 h post each HDM atomization. On day 28, mice were subjected to the examination of methacholine-provoked airway hyperreactivity and euthanized by intraperitoneal injection of excess pentobarbital sodium for sampling.

### A mouse model with IL-13-induced airway goblet cell metaplasia

Mice at 8 weeks of age were anesthetized with 2.5% isofluraner (Sigma), and then placed on the restraining device in a vertical head-up position with the hind legs on the bench. Two front teeth were fixed in a loop of string, and the tongue was gently pulled out to prevent any swallowing. On day 0, 2, 4 and 6, mice were intratracheally administered with either 50 μl/mouse of 2-D08 at the dosages of 10 and 30 mg/kg or vehicle. On day 1, 3, 5, mice were intratracheally administered with either 50 μl/mouse of 4 ug recombinant mouse IL-13 or NS as a control, and kept in this position until all the fluid is inhaled. 24 h after instillation, mice were euthanized by intraperitoneal injection of excess pentobarbital sodium for sampling.

### BALF preparation and cell counting and classification

Preparation of BALFs was performed as previously described[47,48,54]. Briefly, 24 h after the last OVA or HDM challenge, mice were anesthetized with urethane (2 g/kg, i.p.), and BALFs were obtained via tracheal tube and washing the lung with 0.4 ml of sterilized NS containing 1% bovine serum albumin (BSA) three times. BALFs were centrifuged at 800 g at 4 °C for 10 min and the pellets were resuspended with Hank's balanced salt solution (HBSS) for cell counting, Wright-Giemsa staining and classification.

### H&E and PAS staining and immunostaining

Histopathological examination was performed as previously described[55]. Briefly, the partial left lower lobe of lungs was infused via trachea with 10% neutralized formalin for 7 days. After lungs were paraffinized, 5 μm sections were subject to H&E and PAS staining. PAS staining was performed by using kit (Sigma). Immunohistochemistry

staining was performed by using a Histostain-Plus Kit (Kangwei Reagents, Beijing, China) as described previously[56]. Primary antibodies against p-S1366-ROCK2 (ab228008, Abcam, Cambridge, UK, 1:100), ROCK2 (ab228000, Abcam, 1:100), SAE1 (ab185552, Abcam, 1:100), SAE2 (ab185955, Abcam, 1:100), UBC9 (ab75854, Abcam, 1:100), PIAS1 (ab109388, Abcam, 1:50) were incubated at 4 °C overnight. Immunofluorescence staining was performed as described previously[57]. Lung sections were incubated with primary antibody at 4 °C overnight followed by washing, and then further incubated with Alexa555 or Alexa488-conjugated secondary antibody (Invitrogen, Grand Island, NY, 1:1000). Primary antibodies used were as follows: Muc5AC (ab3649, Abcam, 1:50), CC10 (sc365992, Santa Cruz Biotechnology, Santa Cruz, CA, 1:200), Samples were captured by Olympus confocal fluorescence microscope. TIFF files were obtained and semi-quantitative histomorphometry for PAS, immunohistochemistry and immunofluorescence staining was performed blindly in six different fields for each sample by using Image-Pro Plus 6.0 software (Media Cybernetics, Silver Spring, MD). Parameters including area sum, integrated optical density (IOD), and number of positive cells were assessed.

## Determination of airway hyperreactivity

Twenty-four hours after the last antigen challenge, mice were anesthetized and tracheotomized, and the airway resistance was measured by using AniRes 2005 animal lung function analysis (Bestlab Technology Co, Beijing, China) as described previously[58,59]. Mice were challenged with 50 μl of methacholine (Mch, Sigma, dissolved in NS) in gradient concentrations of 0, 0.0125, 0.025, 0.05, 0.1, 0.2, 0.4, 0.8, and 1.6 mg/kg via intra-jugular administration. Airway responsiveness was assessed by the AniRes2005 software with the parameters including Mch volume 50 μl, aerosol period 3 ~ 4 s, determining duration 5 min, which were based on the criterion that the resistance curves must return to the pre-Mch level before the next Mch injection. The results were measured as the peak increase above the baseline immediately after Mch administration.

## Isolation of mouse primary bronchial epithelial cells

Mouse primary bronchial epithelial cells were isolated and cultured as described previously[60]. After removal of the surrounding connective tissue and blood vessels, bronchi from C57BL/6J mice cut into small segments and incubated in 1% protease type XIV (Life Technologies, Carlsbad, CA) containing 0.01% deoxyribonuclease I (DNase I) on a shaker at 4 °C for 24 h. The cell suspension was harvested for centrifugation at 800 g for 5 min, and the pellet was incubated in the anti-clumping solution containing 0.25 mg/ml collagen type IV (Life Technologies), 2 mM EDTA, 0.05 mg/ml dithiothreitol (DTT) and 10 μg/ml DNase for 15 ~ 60 min at 37 °C and centrifugation, cells were layered and expanded in bronchial epithelial cell growth medium (BEGM, Lonza, Swiss) containing 10% febal bovine serum (FBS) at 37 °C with 5% $CO_2$.

## Cell cultures and transfection

HEK293T (293T) cells were purchased from ATCC (Manassas, VA) and cultured in Dulbecco's Modified Eagle Medium (DMEM, Life Technologies). Human bronchial epithelial cells, 16HBE cells, were derived from a differentiated SV-40 transformed bronchial epithelial cell line, generously gifted from Professor Huahao Shen from Zhejiang University School of Medicine, and cultured in Roswell Park Memorial Institute-1640 medium (RPMI-1640, Life Technologies). Cells grew in MEM with Earle's salts (Life Technologies) supplemented with 10% FBS. All cell lines were subcultured before reaching confluence and incubated at 37 °C with 5% $CO_2$. Transient transfections were performed by using Lipofectamine 2000 reagent (Life Technologies) as described previously[61]. Cells were transfected with vector, Flag-ROCK2(WT), Flag-ROCK2(K1007R), Myc-SUMO1, Myc-caRhoA,

HA-RhoA or Myc-PIAS1/2/3/4 construct, and further incubated for 24 ~ 48 h before various analyses.

## Cell lysate preparation, co-immunoprecipitation, and western blotting

Cells were washed three times in PBS, and cell lysates were prepared by RIPA lysis buffer. Cytosolic and nuclear fractions of cells were prepared by using NE-PER Nuclear and Cytoplasmic Extraction Reagents (Thermo Scientific). Co-immunoprecipitation experiment was performed in the supernatants of cell lysates, after centrifugation at 13000 g for 15 min. Supernatant was incubated with antibody and protein A/G PLUS-Agarose (Santa Cruz) overnight at 4 °C. The bound proteins were eluted in the loading buffer and subjected to the SDS-PAGE. Total protein extracts from cells or bronchi were prepared in cell lysis buffer, and the same amount of protein was subjected to 8–15% SDS-PAGE. Western blotting was performed by a standard method as described previously[48]. Primary antibodies were used as follows: p-S1366-ROCK2 (ab228008, Abcam, 1:1000), ROCK2 (ab228000, Abcam, 1:2000), SAE1 (ab185552, Abcam, 1:1000), SAE2 (ab185955, Abcam, 1:1000), UBC9 (ab75854, Abcam, 1:1000), p-S19-MLC2 (#3671, CST, Danvers, MA, 1:500), MLC2 (#3672, CST, 1:500), SUMO1 (#4930, CST, 1:1000), Flag (#14793, CST, 1:3000), Myc (#2276, CST, 1:3000), HA (#3724, CST, 1:2000), glyceraldehyde 3-phosphate dehydrogenase (GAPDH, SC-32233, Santa Cruz, 1:3000). IRDye 680 and 800 secondary antibodies (LI-COR Biosciences, Lincoln, NE, 1:10,000) were used for detecting the immunoreactive bands, and immunoreactive signals were visualized by Odyssey Infrared Imaging System (LI-COR Biosciences). Semi-quantification was performed by using ImageJ (NIH, Bethesda, MD). The phosphorylated protein was normalized to its total protein, and total protein was normalized to GAPDH.

## Quantitative RT-PCR

Total RNA was isolated by using a Trizol reagent (Takara Biotechnology Co., Ltd., Dalian, China) as per manufacturer's instruction. 2 μg total RNA was reversely transcribed by using SuperScript III reagent (Life Technologies). Messenger RNA levels of interest genes were determined by qPCR as previously described[55,62]. The relative amounts of each mRNA level were normalized to GAPDH levels, and the differences in mRNA levels were calculated by $2^{-\Delta\Delta Ct}$. The sequences of used primers are summarized in Supplementary Table 1.

## In vitro SUMOylation and GTP-RhoA pull-down assays

In vitro SUMOylation assays was performed as described previously[63]. Expression of ROCK2(WT) and ROCK2(K1007R) recombinant proteins was performed by using pBAD in TOP 10 F' Escherichia coli as per manufacturer's instructions (Thermo Scientific). TOP 10 F' recombinant protein expression was induced with L-arabinose followed by purification of the His-tagged proteins with a $Ni^{2+}$-nitrile-acetic acid-agarose affinity column, concentrated by a centrifugal filter, and dissolved in a buffer containing 10 mM Tris (pH 7.0) and 1 mM EDTA. In vitro SUMOylation assay was performed with an in vitro SUMOylation kit (Enzo Life Sciences, Farmingdale, NY) as per manufacturer's protocol. Purified ROCK2(WT) and ROCK2(K1007R) proteins were incubated in a SUMOylation buffer containing E1, E2, and SUMO-1 in the presence of $Mg^{2+}$-ATP at 4 °C or 37 °C for 60 min, after termination, proteins were subjected to SDS-PAGE followed by western analyses.

RhoA activation assays were performed by using an EZ-Detect RhoA Activation Kit (Thermo Scientific) as described previously[48]. Active form of Rho (GTP-Rho) was pulled down by a GST fusion protein containing the Rho binding domain of Rhotekin (GST-Rhotekin RBD). The active form of Rho and ROCK2 proteins were detected by immunoblotting using anti-RhoA and anti-p-ROCK2 antibodies.

## Statistical analyses

Numerical data are expressed as means ± SD. Statistics were performed by using Student's $t$ test or One-way ANOVA and Tukey-Kramer multiple comparisons test (SPSS 26.0, SPSS Inc., Chicago, IL). Statistical significance was assessed at levels of $P < 0.05$ and $P < 0.01$. Experiments were repeated at least three times with similar results, and representative results were shown.

## Reporting summary

Further information on research design is available in the Nature Portfolio Reporting Summary linked to this article.

## Data availability

The quantitative phosphoproteomics data has been deposited to the iProX database with the accession code IPX0006541000 (https://www.iprox.cn//page/project.html) The data that support this study are available within the article and its Supplementary Information files. Source data are provided with this paper.

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

## Acknowledgements

This work was supported by the National Natural Science Foundation of China (Nos. 82070027 M.L., 32170841 X.Wu., 31871395 X.Wu., 82000046 C.X.) and Science and Technology Department of Zhejiang Province (LQ21H010001 C.X., LY22H150002 J.W., GF22H155752 F.W.).

## Author contributions

D.T., M.L., C.X., Y.C., H.B., and X.Wu. conducted experiments and analyzed the data and wrote the manuscript. W.Q., F.W., X.Wang., L.T., and J.W. collected the clinical samples. M.Q., Q.H., Y.X., Y.H., T.S., J.L., and M.Q. contributed to discussion and reviewed and edited the manuscript. M.L. and X.Wu. are the guarantors of this work.

## Competing interests

The authors declare no competing interests.

## Additional information

[1]Department of Pharmacology, Zhejiang University School of Medicine, Hangzhou 310058, China. [2]Key Laboratory of CFDA for Respiratory Drug Research, Zhejiang University School of Medicine, Hangzhou 310058, China. [3]National Clinical Research Center for Child Health, the Children's Hospital of Zhejiang University School of Medicine, Hangzhou 310053, China. [4]Department of Thoracic Surgery, the Affiliated Hospital of Jiaxing University, Jiaxing 314001, China. [5]Department of Paediatrics, the First People's Hospital of Wenling City, Wenling City 317500, China. [6]Department of Critical Care Medicine, the First Affiliated Hospital, Zhejiang University School of Medicine, Hangzhou 310003, China. [7]These authors contributed equally: Dan Tan, Meiping Lu. ✉e-mail: meipinglu@zju.edu.cn; xuchengyun@zju.edu.cn; xiwu@zju.edu.cn

