## [Peer Review File · Nature Communications]

SUMOylation of Rho-associated protein kinase 2 induces goblet cell metaplasia in allergic airwaysREVIEWER COMMENTS

Reviewer #1 (Remarks to the Author):

In this manuscript, Tan et al. investigate the role and underlying mechanism of protein Sumoylation contributes to allergic airway goblet cell metaplasia. By phosphoproteomics and biochemical analyses, they identify that ROCK2 can be SUMOylated on K1007 by E3 ligase PIAS1. They show that SUMOylation-mediated activation of ROCK2 enhanced the binding of ROCK2 and RhoA. More importantly, they provide the evidences in mice model that SUMOylation deficiency ROCK2 attenuates allergic airway goblet cell metaplasia. Overall, the main finding of this work is of interest. Nevertheless, there are both major and minor issues the authors should address before this manuscript is ready for publication.

Major issues:

1. Although the authors show the effect of OVA or IL-13 on the components of SUMOylation machinery in 16HBE cells and whole bronchi of mice, they performed co-immunoprecipitation experiments only in 293T cells. They should confirm the co-immunoprecipitation results in other cell types, preferentially human or mice primary cells.
2. To be more convincing, given that SUMOylation of ROCK2 enhance the binding of ROCK2 and RhoA, it will help strengthen the point if the authors perform in vivo experiments that SUMO inhibitor treatment can affect the co-localization of ROCK2 and RhoA in mice lung epithelial cells with immunofluorescence staining.
3. In Figure 5C, the authors show the interaction of ROCK2 and SUMO1 by co-immunoprecipitation assay. A weakness of this experiment is that no difference between the blotting of immunoprecipitation and input samples, indicating the poor immunoprecipitation efficiency.

Minor issues:

1. In Figure 1A and Figure 6G, the authors show the immunostaining of the SUMOylation machinery components in human bronchial tissues and claimed that they are high expression in these tissues. However, based on the principal of the immunostaining methods, the authors cannot draw this conclusion without any control. They should either add appropriate control tissues or remove these data.
2. In Figure 4A, the schematic diagram of mice study is confusing and inconsistent with the figure legend and result.
3. In Figure 6D, the protein level of Flag-ROCK2 is not equal in input.

Reviewer #2 (Remarks to the Author):

In this manuscript, the authors have investigated the role of protein SUMOylation in allergic asthma. Phosphoproteomics identified the ROCK2 kinase as a major phosphorylated protein following stimulation with IL-13, an inducer of goblet cell metaplasia, which could be significantly reduced by treatment with the UBC9 inhibitor 2-D08. Biochemical analysis identified K1007 as the site of SUMOylation, mutation of this site to an arginine residue attenuated SUMOylation and activation as assayed by autophosphorylation on Ser1366. A knock-in ROCK2 K1007R mouse was resistant to allergen-induced airway inflammation, goblet cell metaplasia and hyper-reactivity, as well as IL-13-induced goblet cell metaplasia, supporting the central role of ROCK2 SUMOylation on K1007 in this pathological condition.

The experiments are largely convincing, and the conclusions are compelling. The mechanism of ROCK2 activation is largely credible, but there is a concern that the read-out of Ser1366 phosphorylation as a marker of ROCK2 activation does not necessarily indicate that there will be parallel increases in substrate phosphorylation, especially of myosin light chain (MLC). The conclusions would be more strongly supported if additional blotting for pMLC vs total MLC were included along with the pS1366 blotting of ROCK2, to show that there is a correlation between kinase autophosphorylation and substrate phosphorylation.

The proposed mechanism of activation via increased association with RhoA in Figure 5L is not convincing. When cells expressed wild-type FLAG-tagged ROCK2 or K1007R ROCK2, along with myc-tagged active RhoA, were lysed and proteins immunoprecipitated with anti-FLAG antibody, a anti-myc antibody reactive band is visible at 180 kDa, which could be the size of ROCK2 + RhoA. But this interaction is not covalent and would be expected to be disrupted by solubilization in sample buffer and PAGE. What would be expected, if the proposed mechanism were correct, would be a myc-tagged RhoA band at about 21 kDa. Can the authors explain this observation?

In fact, it would be more convincing if IPed ROCK2 co-purified with increased levels of normal RhoA (not constitutively active) after conditions that induce ROCK2 SUMOylation, such as IL-13 treatment. Furthermore, given that Figure 5E demonstrates that ROCK2 can be SUMOylated in vitro, a cell-free experiment in which the ability of recombinant GTP-loaded RhoA were tested for binding to wild-type vs K1007R ROCK2 subjected to comparable in vitro SUMOylation would strongly support the proposed model.

Reviewer #3 (Remarks to the Author):

Comments to the Author

General comments:

The authors assessed SUMOylation using human samples, murine models of allergic airway responses, and cell lines, and found that SUMOylation-mediated ROCK2 activation was a critical component of Rho/ROCK signaling pathway and might play a determinant role in controlling the airway goblet cell metaplasia and most likely the overall features of allergic airway responses. They concluded that blockage of this SUMOylation pathway might be a promising or additional approach for therapeutic intervention of allergic asthma. Overall, this study is well organized, but in order to investigate the pathogenesis of asthma, this study has some limitations in assessment of the murine model of severe asthma.

Specific comments:

1. The authors utilized an OVA-sensitized and challenged model, which is classical and mostly dependent on Th2 cells but not innate lymphoid cell type 2 (ILC2) cells. Many recent studies have reported that ILC2 cells are involved in the pathogenesis of asthma including severe asthma, in which these studies have assessed ILC2 involved models using house dust mite or fungus instead of OVA as an allergen, or murine models of asthma using IL-33, TSLP, or papain. In the present study, assessment of SUMOylation-mediated ROCK2 activation in one of above models is recommended.

2. In Figure 1A, SAE1, SAE2, and UBC9 are stained, however, it is difficult to detect on which cell types they are expressed.

3. Airway hyperresponsiveness is the most important phenotype of bronchial asthma. In the present study, the authors assessed it indirectly using Penh but not direct measurement of airway resistance, which is a critical limitation of this study. At least, I recommend that a further analysis on this point be included in the discussion part.

Response to the respected reviewers

We wish to express our appreciation for reviewer's valuable comments that improved the quality of our manuscript. We have made our effort to address the critiques thoroughly in this revision. Our response to the reviewers' specific comments and changes made during the revision are detailed below. We again appreciate your helpful suggestions. If you have any further suggestions for changes, please let us know.

Reviewer #1

Comment #1: In this manuscript, Tan et al. investigate the role and underlying mechanism of protein Sumoylation contributes to allergic airway goblet cell metaplasia. By phosphoproteomics and biochemical analyses, they identify that ROCK2 can be SUMOylated on K1007 by E3 ligase PIAS1. They show that SUMOylation-mediated activation of ROCK2 enhanced the binding of ROCK2 and RhoA. More importantly, they provide the evidence in mice model that SUMOylation deficiency ROCK2 attenuates allergic airway goblet cell metaplasia. Overall, the main finding of this work is of interest. Nevertheless, there are both major and minor issues the authors should address before this manuscript is ready for publication.

Response #1: We greatly appreciate the reviewer's professional and insightful comments that really improve the quality of manuscript. We have performed additional experiments and addressed almost all issues raised by the reviewer.

Comments #2: Although the authors show the effect of OVA or IL-13 on the components of SUMOylation machinery in 16HBE cells and whole bronchi of mice, they performed co-immunoprecipitation experiments only in 293T cells. They should confirm the co-immunoprecipitation results in other cell types, preferentially human or mice primary cells.

Response #2: We agree with the reviewer's suggestion and have performed the

additional experiments in 16HBE cells and mouse primary bronchial epithelial cells. Additional co-immunoprecipitation experiments were performed in 16HBE cells and mouse primary bronchial epithelial cells (**Fig. 5c**) and indicated SUMO1 interacted with ROCK2 in both 16HBE cells and mouse primary bronchial epithelial cells. Additional co-immunoprecipitation experiments were performed in 16HBE cells and indicated Ubc9 knockdown attenuated ROCK2 SUMOylation (**Fig. 5d**). Additional co-immunoprecipitation experiments were performed in 16HBE cells and indicated immunocomplex precipitated by a Flag antibody contained abundant Myc-SUMO1-conjugated ROCK2(WT) but rare Myc-SUMO1-conjugated ROCK2(K1007R) in 16HBE cells expressing Myc-SUMO1 and Flag-ROCK2 variants (**Fig. 5g**).

Comments #3: To be more convincing, given that SUMOylation of ROCK2 enhance the binding of ROCK2 and RhoA, it will help strengthen the point if the authors perform in vivo experiments that SUMO inhibitor treatment can affect the co-localization of ROCK2 and RhoA in mice lung epithelial cells with immunofluorescence staining.

Response #3: We have now performed the additional experiments accordingly and showed that 2-D08, a specific UBC9 inhibitor, robustly decreased the co-localization of p-ROCK2 and RhoA in bronchial epithelia of mice challenged with allergen (**Supplementary Fig. 6**).

Comments #4: In Figure 5C, the authors show the interaction of ROCK2 and SUMO1 by co-immunoprecipitation assay. A weakness of this experiment is that no difference between the blotting of immunoprecipitation and input samples, indicating the poor immunoprecipitation efficiency.

Response #4: We agree with the reviewer and have modified the experiments to improve immunoprecipitation efficiency (**Supplementary Fig. 4a**).

Comments #5: In Figure 1A and Figure 6G, the authors show the immunostaining of the SUMOylation machinery components in human bronchial tissues and claimed that

they are high expression in these tissues. However, based on the principal of the immunostaining methods, the authors cannot draw this conclusion without any control. They should either add appropriate control tissues or remove these data.

Response #5: We have now presented the images with higher magnification and also showed the negative controls (IgG instead of specific antibodies) in the immunochemistry experiments (**Fig. 1a** and **Fig. 6g**).

Comments #6: In Figure 4A, the schematic diagram of mice study is confusing and inconsistent with the figure legend and result.

Response #6: We apologize for the writing mistake and have corrected the scheme in the current version of manuscript (**Fig. 4a**).

Comments #7: In Figure 6D, the protein level of Flag-ROCK2 is not equal in input.

Response #7: We have repeated the experiments and replaced the blotting images by more appropriate ones, where protein levels of Flag-ROCK2 variants were equal in input (**Fig. 6d**).

Reviewer #2

Comments #1: In this manuscript, the authors have investigated the role of protein SUMOylation in allergic asthma. Phosphoproteomics identified the ROCK2 kinase as a major phosphorylated protein following stimulation with IL-13, an inducer of goblet cell metaplasia, which could be significantly reduced by treatment with the UBC9 inhibitor 2-D08. Biochemical analysis identified K1007 as the site of SUMOylation, mutation of this site to an arginine residue attenuated SUMOylation and activation as assayed by autophosphorylation on Ser1366. A knock-in ROCK2 K1007R mouse was resistant to allergen-induced airway inflammation, goblet cell metaplasia and hyper-reactivity, as well as IL-13-induced goblet cell metaplasia, supporting the central role of ROCK2 SUMOylation on K1007 in this pathological condition. The experiments are largely convincing, and the conclusions are compelling. The mechanism of ROCK2 activation is largely credible, but there is a concern that the read-out of

Ser1366 phosphorylation as a marker of ROCK2 activation does not necessarily indicate that there will be parallel increases in substrate phosphorylation, especially of myosin light chain (MLC). The conclusions would be more strongly supported if additional blotting for pMLC vs total MLC were included along with the pS1366 blotting of ROCK2, to show that there is a correlation between kinase autophosphorylation and substrate phosphorylation.

Response #1: We greatly appreciate the reviewer's valuable and constructive comments that help improve the manuscript. We have now performed the additional experiments of p-MLC2 versus MLC2 (**Supplementary Fig. 2g, Fig. 3d, Fig. 5a, Fig. 5b, Fig. 5f, and Fig. 6p**). These results indicate the p-Ser1366-ROCK2 levels are parallel to p-MLC2 levels and p-Ser1366-ROCK2 is capable to serve as a marker of ROCK2 activation.

Comments #2: The proposed mechanism of activation via increased association with RhoA in Figure 5L is not convincing. When cells expressed wild-type FLAG-tagged ROCK2 or K1007R ROCK2, along with myc-tagged active RhoA, were lysed and proteins immunoprecipitated with anti-FLAG antibody, an anti-myc antibody reactive band is visible at 180 kDa, which could be the size of ROCK2 + RhoA. But this interaction is not covalent and would be expected to be disrupted by solubilization in sample buffer and PAGE. What would be expected, if the proposed mechanism were correct, would be a myc-tagged RhoA band at about 21 kDa. Can the authors explain this observation?

Response #2: We completely agree with the reviewer. Because the 21 kDa RhoA bands were accidentally cut off in the previous version of manuscript, we have redone the experiments, where the HA-tagged RhoA was used instead of Myc-tagged active form of RhoA (**Fig. 5I**). Results indicate a HA-tagged RhoA bands at 21 kDa is readily detected along with the weak 180 kDa bands in 16HBE cells expressing FLAG-tagged ROCK2(WT or K1007R) and HA-RhoA (**Fig. 5I**).

Comments #3: In fact, it would be more convincing if IPed ROCK2 co-purified with

increased levels of normal RhoA (not constitutively active) after conditions that induce ROCK2 SUMOylation, such as IL-13 treatment.

Response #3: We have now performed additional co-immunoprecipitation experiments accordingly. Protein complexes precipitated with a Flag antibody and blotted with a HA antibody contained abundant Flag-ROCK2(WT) but much little Flag-ROCK2(K1007R) in 16HBE cells expressing HA-RhoA and Flag-ROCK2 variants, especially in the presence of IL-13 stimulation (**Fig. 5l**).

Comments #4: Furthermore, given that Figure 5E demonstrates that ROCK2 can be SUMOylated in vitro, a cell-free experiment in which the ability of recombinant GTP-loaded RhoA were tested for binding to wild-type vs K1007R ROCK2 subjected to comparable in vitro SUMOylation would strongly support the proposed model.

Response #4: We have now performed additional GTP-RhoA pull-down assays to test the hypothesis. As expected, endogenous GTP-RhoA interacted with and activated much more Flag-ROCK2(WT) than Flag-ROCK2(K1007R) (**Fig. 5m**), suggesting that SUMOylation of ROCK2 indeed affects its binding to and activation by an active form of RhoA (GTP-RhoA).

Reviewer #3

Comments #1: The authors assessed SUMOylation using human samples, murine models of allergic airway responses, and cell lines, and found that SUMOylation-mediated ROCK2 activation was a critical component of Rho/ROCK signaling pathway and might play a determinant role in controlling the airway goblet cell metaplasia and most likely the overall features of allergic airway responses. They concluded that blockage of this SUMOylation pathway might be a promising or additional approach for therapeutic intervention of allergic asthma. Overall, this study is well organized, but in order to investigate the pathogenesis of asthma, this study has some limitations in assessment of the murine model of severe asthma.

Response #1: We are grateful to the reviewer for the constructive and valuable comments that do improve the quality of manuscript. According to the reviewer's

suggestion, we have now performed additional experiments to strengthen this study. Especially, we have assessed the house dust mite (HDM)-induced mouse model of asthma to enrich the study of asthma model.

Comments #2: The authors utilized an OVA-sensitized and challenged model, which is classical and mostly dependent on Th2 cells but not innate lymphoid cell type 2 (ILC2) cells. Many recent studies have reported that ILC2 cells are involved in the pathogenesis of asthma including severe asthma, in which these studies have assessed ILC2 involved models using house dust mite or fungus instead of OVA as an allergen, or murine models of asthma using IL-33, TSLP, or papain. In the present study, assessment of SUMOylation-mediated ROCK2 activation in one of above models is recommended.

Response #2: We completely agree with the reviewer and have now performed additional experiments of house dust mite (HDM)-induced mouse model of asthma (**Supplementary Fig. 2**) to strengthen the study. Results indicate SUMOylation of ROCK2 in HDM-induced asthma mouse model plays essentially the same role as in OVA-induced asthma mouse model (**Supplementary Fig. 2**).

Comments #3: In Figure 1A, SAE1, SAE2, and UBC9 are stained, however, it is difficult to detect on which cell types they are expressed.

Response #3: In the current version of manuscript, we have presented the images with higher magnification and distinctly shown that SAE1, SAE2, UBC9, and PIAS1 are specifically and robustly expressed in the airway epithelia (**Fig. 1a** and **Fig 6g**). In addition, we have used the IgG as a negative control to determine the specificity of antibodies in the immunostaining.

Comments #4: Airway hyperresponsiveness is the most important phenotype of bronchial asthma. In the present study, the authors assessed it indirectly using Penh but not direct measurement of airway resistance, which is a critical limitation of this study. At least, I recommend that a further analysis on this point be included in the

discussion part.

Response #4: We thank the reviewer for the professional and insightful comments. In the past few years, our laboratory had used PenH to assess the lung function in asthma mouse models. However, in recent years, we have used airway resistance to assess the lung function (Wang X, Xu C, Wu X, et al. CircZNF652 promotes the goblet cell metaplasia by targeting the miR-452-5p/JAK2 signaling pathway in allergic airway epithelia. *J Allergy Clin Immunol.*2022 Jul;150(1):192-203). We actually assessed the lung function by measuring the airway resistance in the present study and miswrote the method of airway hyperreactivity determination in the previous version of manuscript. We have now rearranged the experiments and corrected the description of method (**Fig. 2f, Fig. 7f and Supplementary Fig. 2f, Page 29, Line 558-569**).

REVIEWERS' COMMENTS

Reviewer #1 (Remarks to the Author):

I am satisfied with this revised version. There are no further concerns about this manuscript.

Reviewer #2 (Remarks to the Author):

I am satisfied with the revisions to this manuscript and recommend it be accepted.

Reviewer #3 (Remarks to the Author):

The manuscript has been revised well.